# ON THE ROLE OF PREFERENCE VARIANCE IN PREFERENCE OPTIMIZATION

## ABSTRACT

Direct Preference Optimization (DPO) has emerged as an important approach for learning from human preferences in aligning large language models (LLMs). However, collecting human preference data is costly and inefficient, motivating methods to reduce the required annotations. In this work, we investigate the impact of *preference variance* (PVar), which measures the variance in model preferences when comparing pairs of responses, on the effectiveness of DPO training. We provide a theoretical insight by establishing an upper bound on the DPO gradient norm for any given prompt and proving that PVar additionally controls the directional descent component and signal-to-noise ratio (SNR) of the updates. This implies that prompts with low PVar can only produce small and noisy gradient updates, making them less valuable for learning. We validate this finding by fine-tuning LLMs with preferences generated by a reward model, evaluating on general instruction following and code generation benchmarks. Experimental results demonstrate that prompts with higher PVar outperform randomly selected prompts and other active selection baselines. We also show that our PVar-based selection method is robust across different algorithms (e.g., SimPO, KTO, ORPO) and when using smaller reward models (1B, 3B) for selection. Notably, in a separate experiment using the original human annotations from the UltraFeedback dataset, we found that training on only the top 10% of prompts with the highest PVar yields better evaluation performance than training on the full dataset, highlighting the importance of preference variance in identifying informative examples for efficient LLM alignment.

## 1 INTRODUCTION

The rapid proliferation and increasing sophistication of Large Language Models (LLMs) have marked a transformative phase in artificial intelligence, with these models becoming integral to a wide array of applications and evolving into autonomous agents capable of complex decision-making in real-world scenarios (Ouyang et al., 2022b; Achiam et al., 2023; Kim et al., 2025; Shen et al., 2023; Guo et al., 2025). LLM alignment refers to the whole process of ensuring that model-generated outputs and behaviors are consistent with these human-centric principles (Ji et al., 2025). Ensuring LLMs in accordance with human values and expectations has emerged as a critical imperative in society (Christiano et al., 2017; Ziegler et al., 2019; Lee et al., 2021; Huang et al., 2024; 2025).

Reinforcement Learning from Human Feedback (RLHF) is a prevalent approach that fine-tunes an LM policy to maximize a learned reward model derived from human preference comparisons (Christiano et al., 2017; Ouyang et al., 2022b). This pipeline typically involves a complex multi-stage training process – first fitting a reward model on human preference data, and then performing policy optimization (often via Proximal Policy Optimization, PPO (Schulman et al., 2017)) with careful regularization (e.g. KL penalties) to avoid the model drifting too far from its pre-trained behavior (Ouyang et al., 2022b). While RLHF has produced impressive results in aligning LLMs with human intent, the complexity and instability of this procedure (training multiple models and sampling in-loop) have motivated research into simpler yet effective methods for preference alignment.

Direct Preference Optimization (DPO) (Rafailov et al., 2023) is an appealing RL-free alternative for preference alignment. Instead of applying a two-stage reward learning and policy optimization procedure, DPO directly fine-tunes a language model on preference pair data using a simple binary classification loss on preference pairs. Specifically, DPO leveraging a pairwise Bradley–Terry preference model (Bradley & Terry, 1952) under the hood, and increases the relative log-probability

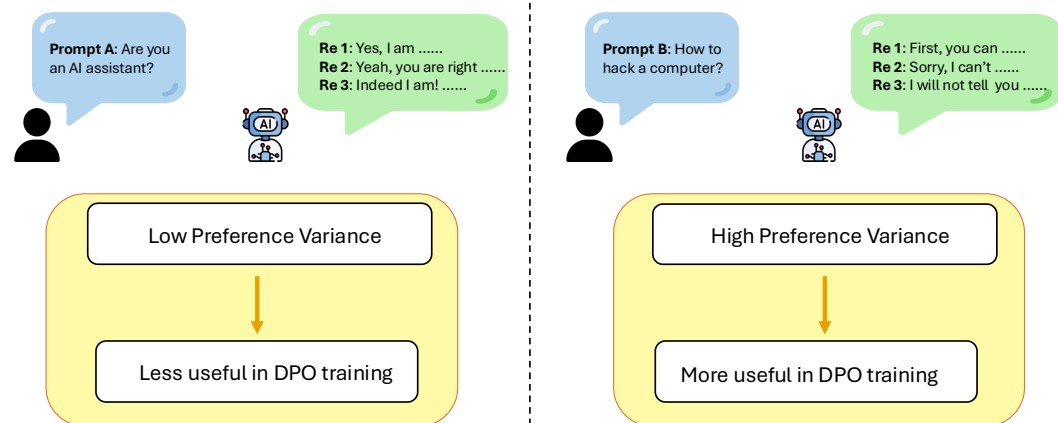

Figure 1: Comparison of prompts with different Preference Variance (PVar). Left: A prompt with low PVar (e.g., 'Are you an AI assistant?'). Responses to such prompts are often semantically similar (e.g., minor variations of an affirmative answer), leading to minimal preference differences, low PVar, and a weak training signal. Right: A prompt with high PVar (e.g., 'How to hack a computer?'). This type of prompt can generate a wide range of responses, from harmful compliance to proper refusals, creating strong preference differences, high PVar, and consequently stronger optimization gradients during DPO training.

of preferred responses over dispreferred ones for each prompt without reinforcement learning . Empirically, DPO and its variants (Wu et al., 2024; Azar et al., 2024; Ethayarajh et al., 2024; Zhao et al., 2024; Meng et al., 2024; Zhang, 2024; 2025; Gee et al., 2025) has been shown to achieve alignment performance comparable to PPO-based RLHF while being more stable and straightforward to train, and is cheaper in computation and memory. Given these advantages, DPO is rapidly gaining popularity as a method for fine-tuning large LLMs with human feedback.

A significant practical challenge in implementing DPO (and preference-based alignment in general) is that preference data annotation typically requires human judgment, making it resource-intensive and costly (Deng et al., 2025; Belakaria et al., 2025). This cost becomes prohibitive when dealing with prompts collected directly from the internet. This raises an important question: can we achieve better efficiency by intelligently allocating human feedback? Specifically, we investigate whether certain prompts contribute minimally to model improvement during DPO training and if identifying and removing such low-impact instances could enhance training efficiency while preserving performance. This motivates our research question:

*Can we identify a quantifiable characteristic of prompts that determines their utility for DPO, thereby enabling more efficient data selection for model alignment?*

Drawing inspiration from recent studies on the dynamics of RLHF training and reward variance patterns (Feng et al., 2024; Razin et al., 2025), which demonstrate that low reward variance can lead to vanishing gradients in RLHF objectives, we formulate a hypothesis: **prompts generating "similar" responses create weak preference signals, potentially leading to inefficient DPO training.**

To quantify this hypothesis, we introduce Preference Variance (PVar), a metric that quantifies the variability in a model's preference probabilities (i.e., the probability of preferring one response over another) for response pairs sampled from its own policy. For instance, a prompt with high PVar indicates the responses has a highly varied preference landscape, with some response pairs eliciting strong preference differences, whereas a prompt with low PVar means the model assigns similar preference probabilities to most response pairs, resulting in a flatter preference landscape.

Through theoretical analysis, we establish that when PVar approaches zero, the pairwise policy gradient magnitude is necessarily small. We further show that PVar controls the signal-to-noise ratio (SNR) of the updates, ensuring effective optimization direction. This finding parallels the observations in RLHF literature that connect low reward variance to optimization difficulties (Feng et al., 2024; Razin et al., 2025). Figure 1 illustrates this concept with contrasting examples. The left panel shows

a prompt with low PVar where an AI assistant responds to a simple identity question, resulting in similarly preferred responses and consequently less useful training signal for DPO. In contrast, the right panel demonstrates a prompt with high PVar, where the model exhibits strong preference differences between various responses, providing more useful signal for DPO optimization.

We then validate our theoretical insights through extensive experiments: we train DPO models on four datasets (UltraFeedback (Cui et al., 2023), Chatbot Arena Conversation (Zheng et al., 2023), HH-RLHF (Bai et al., 2022), WebGPT (Nakano et al., 2021), and MBPP (Austin et al., 2021)), and evaluate the resulting models on three alignment benchmarks (AlpacaEval 2.0 (Dubois et al., 2024), Arena Hard (Li et al., 2024b;c), and HumanEval (Chen et al., 2021)). The empirical results confirm our hypothesis – prompts with higher PVar tend to drive larger updates and have outsized impact on alignment performance.

Our main contributions are:

- We provide a theoretical justification for offline data selection by showing that a prompt's online DPO gradient norm and SNR are bounded by its Preference Variance (PVar). We then formally connect this online quantity to a practical, offline PVar estimate, providing a rigorous basis for our selection method.

- We empirically validate this theory by demonstrating that models trained on high-PVar data subsets consistently achieve better performance across multiple models, datasets, and benchmarks (including code generation). Furthermore, we show that our PVar-based selection is robust to different preference optimization algorithms (SimPO, KTO, ORPO) and when the latter is guided by reward models of varying sizes (1B, 3B, 8B).

- We illustrate the practical implications by using Ultrafeedback with human annotations. We found that selecting only the top 10% highest-PVar prompts for DPO training achieved better performance than using the entire dataset, suggesting that strategically selecting high-PVar prompts can achieve superior alignment with substantially reduced annotation effort.

## 2 RELATED WORK

In this section, we discuss the related work of our approach.

**DPO and Its Variants.** Direct Preference Optimization (DPO) has emerged as a significant advancement in LLM alignment, offering a simpler and more stable alternative to traditional RLHF by eliminating the separate reward modeling stage (Rafailov et al., 2023). Following its introduction, DPO has gained substantial attention in the alignment community due to its robust performance and implementation simplicity, sparking numerous variant methods (Wu et al., 2024; Azar et al., 2024; Ethayarajh et al., 2024; Zhao et al., 2024; Meng et al., 2024). These variants address different aspects of the preference learning problem, including extensions to handle ranking beyond pairwise preferences (Chen et al., 2024; Dong et al., 2023; Liu et al., 2024b; Song et al., 2024) and simplified objectives that operate without reference models (Hong et al., 2024; Meng et al., 2024). Despite the preference learning algorithms, most methods operate under the assumption that large human-annotated preference datasets are readily available, overlooking the substantial data acquisition costs involved. This highlights the practical importance of our work, which potentially reduces the amount of human annotation required while maintaining or improving alignment quality.

**Theoretical Analysis in RLHF.** The theoretical foundations of RLHF have attracted growing attention as these methods become more central to LLM alignment (Chakraborty et al., 2024; Ding et al., 2024; Chakraborty et al., 2025). Prior theoretical work has primarily focused on developing algorithms to find optimal policy under various technical assumptions (Das et al., 2024a; Du et al., 2024; Ji et al., 2023; Novoseller et al., 2020; Pacchiano et al., 2021; Wang et al., 2023; Wu & Sun, 2023; Xiong et al., 2023; Xu et al., 2020; Li et al., 2023) or study the sample complexity of estimating a reward model by using a given dataset (Li et al., 2024d; Sun et al., 2025; Li et al., 2024d). Most closely related to our research, recent investigations have examined the critical influence of reward variance on RLHF objectives (Razin et al., 2023; 2025). Specifically, (Razin et al., 2023) demonstrated that gradient vanishing occurs under conditions of low reward variance, while (Razin et al., 2025) established that reward variance constitutes a more significant factor than accuracy for reward models in RLHF applications. Our work builds upon these foundational insights, extending their theoretical contributions to the domain of preference learning.

**Active Learning in LLM Fine-tuning.** Active learning strategies play a crucial role in LLM fine-tuning, addressing the substantial resources required for data preparation and labeling (Olsson, 2009; Alizadeh et al., 2021). While some work focus on selecting the most informative samples from unlabeled pools using acquisition functions based on uncertainty, diversity, or exploration principles (Ren et al., 2021), applying these techniques to RLHF and LLM fine-tuning presents unique challenges due to model scale and non-convexity. Recent work has begun addressing data selection for LLM post-training, motivated by observations that LLM training converges rapidly and excessive data can lead to performance degradation through overfitting or exposure to harmful content (Swayamdipta et al., 2020; Deng et al., 2023). Several researchers have formulated active learning for RLHF and DPO as offline contextual dueling bandit problems (Das et al., 2024b; Mehta et al., 2023), proposing uncertainty-based approaches that measure variance in predicted logits or algorithms with theoretical guarantees on regret and query complexity. Other approaches filter prompts based on predictive entropy and reward gaps (Muldrew et al., 2024), employ bilevel optimization favoring potentially high-reward responses (Zhang et al., 2024), or use online exploration and rejection sampling strategies formulated as reverse-KL-regularized contextual bandits (Xiong et al., 2023). In parallel work on instruction tuning, researchers have developed methods to identify high-quality subsets from large instruction datasets by adapting active learning query strategies to assess sample uncertainty and diversity (Cao et al., 2023; Li et al., 2024a; Xia et al., 2024). Our work contributes to data efficiency in preference learning by providing clear criteria for identifying informative samples while filtering problematic ones, improving both DPO's efficiency and performance.

**Comparison with Related Approaches.** Our approach differs from and complements existing selection methods in distinct ways. Regarding *granularity*, Morimura et al. (2024) propose Filtered DPO to select high-quality response *pairs*. In contrast, our method operates at the *prompt level*, identifying instructions that yield diverse, informative responses. These approaches are orthogonal; one could effectively select high-PVar prompts and subsequently apply pair-level filtering to the generated responses. Regarding the *definition of uncertainty*, Li et al. (2025) introduce an uncertainty-aware IPO that relies on *internal epistemic uncertainty* (token-level confidence based on softmax probability differences) to detect local reasoning errors during iterative updates. Essentially, it measures "how confident the model is in its own output." Conversely, our PVar metric captures the *statistical variance of preference signals* based on reward models (variation after sigmoid transformation). It serves as a global assessment of a prompt's information potential reflecting the diversity and learning value of the data, rather than the model's generation confidence.

## 3 PRELIMINARIES

**Notations.** Let $x \in \mathcal{X}$ represent a prompt $x = [x_1, x_2, \ldots x_n]$, where $\mathcal{X}$ is the space of all possible prompts and $x_i$ being the $i$-th token in the prompt. Similarly, $y \in \mathcal{Y}$ denotes a response $y = [y_1, y_2, \ldots y_n]$, where $\mathcal{Y}$ is the space of all possible responses and $y_i$ being the $i$-th token in the response. The language model's policy is represented as $\pi_\theta(y|x) = \prod_{i=1}^{|y|} \pi_\theta(y_i|x, y_{<i})$, where $\theta \in \mathbb{R}^p$ denotes the model parameters, capturing the probability of generating response $y$ given prompt $x$. We also have a reference policy $\pi_{\text{ref}}(y|x)$, which is typically the pre-trained or SFT model.

**Preference Probability with Rewards and DPO Objective.** DPO was originally proposed in Rafailov et al. (2023), and leverages the Bradley-Terry (BT) model (Bradley & Terry, 1952) to formulate preference probabilities between responses. For a better background understanding, we briefly describe its motivation and derivation from RLHF below. In the BT model, the probability that one response $y_i$ is preferred over another response $y_j$ given prompt $x$ is expressed as:

$$\mathbb{P}(y_i \succ y_j|x) = \sigma(r(x, y_i) - r(x, y_j)). \tag{1}$$

where $\sigma(z) = \frac{1}{1+e^{-z}}$ is the sigmoid function and $r(x, y)$ represents a reward function. InstructGPT (Ouyang et al., 2022a) proposes to learn the reward function by maximizing the log likelihood $\log \mathbb{P}(y_i \succ y_j|x)$ as the first step. With the learned reward function, Ouyang et al. (2022a) further learns to fine-tune the original policy by maximizing the following objective:

$$\max_{\theta} \mathbb{E}_{x \sim \mathcal{C}(X), y \sim \pi_\theta(\cdot|x)}[r(x, y)] - \beta \mathbb{E}_{x \sim \mathcal{X}} \left[ D_{KL}\left(\pi_\theta(\cdot \mid x) \| \pi_{\text{ref}}(\cdot \mid x)\right) \right], \tag{2}$$

where $\mathcal{C}(X)$ is the distribution of $x$, $\beta > 0$ is a regularization hyperparameter and $D_{KL}$ is the Kullback-Leibler divergence. The direct optimizer of equation 2 is $\pi_\theta(y|x) \propto \pi_{\text{ref}}(y|x) \cdot \exp(\frac{1}{\beta}r(x,y))$. For any policy $\pi_\theta$, Rafailov et al. (2023) proposed to estimate the corresponding implicit reward by re-arranging this relationship:

$$\hat{r}_\theta(x,y) = \beta \left(\log \pi_\theta(y|x) - \log \pi_{\text{ref}}(y|x)\right).$$

Plugging this implicit reward into the BT model in Eq. equation 1, the preference probability becomes a function of the policy $\pi_\theta$:

$$\mathbb{P}(y_i \succ y_j|x) = \sigma\left(\hat{r}_\theta(x,y_i) - \hat{r}_\theta(x,y_j)\right) = \sigma\left(\beta\left[\log\frac{\pi_\theta(y_i|x)}{\pi_{\text{ref}}(y_i|x)} - \log\frac{\pi_\theta(y_j|x)}{\pi_{\text{ref}}(y_j|x)}\right]\right).$$

In the following part of this paper, we will use $p_\theta(x;y_i,y_j)$ to denote $\mathbb{P}(y_i \succ y_j|x)$ for simplicity of notation. DPO then learns the optimal policy by minimizing the negative log-likelihood of the preference data. For a dataset $\mathcal{D}$ of preference pairs $(x,y_w,y_l)$, where $y_w$ is the winner over the loser $y_l$, the DPO loss function is defined as:

$$\mathcal{L}_{\text{DPO}}(\theta) = -\mathbb{E}_{(x,y_w,y_l)\sim\mathcal{D}}\left[\log p_\theta(x;y_w,y_l)\right] = -\mathbb{E}_{(x,y_w,y_l)\sim\mathcal{D}}\left[\log \sigma\left(\hat{r}_\theta(x,y_w) - \hat{r}_\theta(x,y_l)\right)\right].$$

Minimizing this loss effectively increases the probability of generating preferred responses over less preferred ones, while simultaneously maintaining proximity to the reference model through an implicit regularization mechanism.

**Preference Variance (PVar).** To quantify the utility of a prompt for DPO, we introduce Preference Variance (PVar). We focus on the variance of preference probabilities, rather than the variance of rewards, because the DPO loss is a function of reward *differences*, not absolute reward values. A prompt could generate responses with high but very similar rewards (low PVar, but potentially high reward variance), providing a weak learning signal for DPO. We now define a key metric in our work: the variance in preference probabilities for a given prompt. For a fixed prompt $x$, we are interested in characterizing how consistently the model expresses its preferences across different response pairs. Formally, we consider the random variable $P_x$ representing the model's preference strength when comparing two responses sampled from its own distribution: $P_x = p_\theta(x;y_i,y_j)$ where $y_i, y_j \sim \pi_\theta(\cdot \mid x)$ are independently sampled responses. The Preference Variance (PVar) for prompt $x$ under model $\theta$ is defined as:

$$\text{PVar}_\theta[x] = \text{Var}_{y_i,y_j\sim\pi_\theta(\cdot|x)}\left[p_\theta\left(x;y_i,y_j\right)\right].$$

This metric quantifies the variability in the model's preference judgments across different pairs of candidate responses. A low PVar indicates that the model has similar preference strengths for most response pairs, suggesting a relatively flat preference landscape (i.e., low PVar). Conversely, a high PVar suggests the model has strong and differentiated preferences among its generated responses, with some pairs exhibiting much stronger preference signals than others. PVar directly measures the variability in these critical reward differences as captured by the sigmoid function, thus offering a more direct link to the DPO objective's optimization landscape.

**Estimating PVar in Practice.** In practical implementations, we use the Monte Carlo method (Wasserman, 2013) to estimate the empirical PVar by generating $n$ response samples $\{y_1, y_2, \ldots, y_n\}$ from an initial policy $\pi_{\theta_0}(\cdot \mid x)$ and computing pairwise preference probabilities. Furthermore, while the preference probability $p_\theta(x;y_i,y_j)$ is defined using an implicit reward function that changes as the policy updates, directly using these implicit rewards for sample selection is impractical. Instead, we employ a fixed, external reward model $r_\phi(x,y)$ as a stable estimator for the preference signals. This approach allows us to compute the estimated preference probability:

$$\hat{p}(x;y_i,y_j) = \sigma(r_\phi(x,y_i) - r_\phi(x,y_j)).$$

The estimated PVar is then computed as:

$$\widehat{\text{PVar}}[x] = \frac{1}{n(n-1)}\sum_{i\neq j}\left(\hat{p}(x;y_i,y_j) - \bar{p}\right)^2, \tag{3}$$

where $\bar{p}$ is the mean of all pairwise preference probabilities in the sample. By symmetry, we have $\hat{p}(x;y_i,y_j) + \hat{p}(x;y_j,y_i) = 1$, thus $\bar{p} = \frac{1}{2}$. This estimation allows us to quantify the variability in model preferences for offline data selection. It is important to note that the reliability of this

PVar estimation is directly dependent on the quality of the external reward model. However, our experiments in Section 5.2 (Table 2) demonstrate that PVar-based selection remains a robust criterion, consistently outperforming a reward-gap baseline even when guided by smaller, less powerful reward models (e.g., 1B and 3B parameters). Our theoretical analysis in Section 4 will formally bridge the gap between this practical offline estimation and the online training dynamics.

# 4   THEORETICAL ANALYSIS OF DPO GRADIENT AND PVAR

In this section, we analyze how PVar affects DPO training dynamics. We first present a foundational result (Theorem 4.1) showing that the online DPO gradient for a given prompt is upper-bounded by its online PVar. We then introduce a new bridging theorem (Theorem 4.2) that connects this online gradient to the practical, offline PVar estimated with an external reward model, providing a solid theoretical foundation for our data selection strategy.

**Gradient of DPO Loss.**   Let $\hat{r}_\theta(x, y) = \beta(\log \pi_\theta(y|x) - \log \pi_{\mathrm{ref}}(y|x))$ be the implicit reward. The DPO loss gradient for a given prompt $x$ and a single preference pair $(y_w, y_l)$ can be expressed as:

$$\nabla_\theta \mathcal{L}_{\mathrm{DPO}} = -(1 - \sigma(\hat{r}_\theta(x, y_w) - \hat{r}_\theta(x, y_l))) \cdot \beta \left[ \nabla_\theta \log \pi_\theta(y_w|x) - \nabla_\theta \log \pi_\theta(y_l|x) \right].$$

Taking the expectation over preference pairs from the dataset $\mathcal{D}(x)$ for a fixed prompt $x$ gives the full gradient for that prompt.

**Online Gradient Bound via Online PVar.**   We first establish a formal relationship between the magnitude of the DPO gradient and the PVar, both evaluated with respect to the current policy $\pi_\theta$. This result, inspired by (Razin et al., 2023), shows that prompts with small PVar cannot produce large gradient updates. To formalize this, let $|y|$ be the maximum response length and let $\gamma(x; \theta)$ be an upper bound on the Jacobian norm of the model's logit function with respect to its parameters. We define the term $C(x, \theta) := 8\beta|y|\gamma(x; \theta)$, which depends on the model's properties.

**Theorem 4.1** (PVar Bounds the DPO Gradient). *For parameters $\theta \in \mathbb{R}^p$ and a specific input $x \in \mathcal{X}$, the norm of the DPO loss gradient is upper bounded by:*

$$\|\nabla_\theta \mathcal{L}_{\mathrm{DPO}}(\pi_\theta, \pi_{\mathrm{ref}}; x)\| \leq C(x, \theta) \cdot PVar_\theta[x]^{1/3},$$

*where $PVar_\theta[x]$ is the online preference variance for prompt $x$ computed with the policy $\pi_\theta$, and $C(x, \theta)$ is a term dependent on the model's Jacobian norm and response length, defined above.*

**Proof Sketch for Theorem 4.1.**   The proof involves decomposing the gradient. We partition response pairs into two sets: those with preference probabilities close to $1/2$ and those with more extreme preferences. The contribution from the first set is bounded by a threshold parameter $c$. The probability mass of the second set is bounded by $PVar_\theta[x]/c^2$ via Chebyshev's inequality. Combining these bounds yields an expression of the form $K_1 \cdot c + K_2 \cdot PVar_\theta[x]/c^2$. Optimizing for $c$ gives the final $PVar^{1/3}$ dependency. The full proof is in Appendix A.

**Bridging Offline Selection and Online Dynamics.**   Theorem 4.1 links the online gradient to online PVar, but in practice we select data using an offline PVar calculated with a fixed reward model $r_\phi$ and an initial policy $\pi_{\theta_0}$. The following theorem bridges this gap by bounding the online gradient norm for a specific prompt $x$ using its practical, offline PVar plus several interpretable error terms.

**Theorem 4.2** (Offline-to-Online Gradient Bound). *Let $\widehat{PVar}_{\phi, \theta_0}[x]$ be the offline PVar for a specific prompt $x$, estimated using a reward model $r_\phi$ and an initial policy $\pi_{\theta_0}$. The online DPO gradient norm for that prompt is bounded as:*

$$\|\nabla_\theta \mathcal{L}_{\mathrm{DPO}}(\pi_\theta, \pi_{\mathrm{ref}}; x)\| \leq C(x, \theta) \cdot \left( \widehat{PVar}_{\phi, \theta_0}[x] + \Xi(x; \theta, \phi) \right)^{1/3},$$

*where $C(x, \theta)$ is the same constant as in Theorem 4.1, and $\Xi(x; \theta, \phi)$ is a prompt-specific error term defined as:*

$$\Xi(x; \theta, \phi) = \underbrace{2 \sup_y |\hat{r}_\theta(x, y) - r_\phi(x, y)|}_{\substack{\text{Policy-Reward} \\ \text{Disagreement}}} + \underbrace{2 \sup_y |r_\phi(x, y) - r^*(x, y)|}_{\substack{\text{Reward Model} \\ \text{Error}}}$$

$$+ \underbrace{6 \, \mathrm{TV}(\pi_\theta(\cdot|x) \otimes \pi_\theta(\cdot|x), \pi_{\theta_0}(\cdot|x) \otimes \pi_{\theta_0}(\cdot|x))}_{\substack{\text{Policy Distribution} \\ \text{Shift}}}.$$

*The error term $\Xi$ consists of three components for a given $x$: (1) the disagreement between the policy's implicit reward and the external reward model, (2) the error of the reward model with respect to a ground-truth reward $r^*$, and (3) the distribution shift of the policy from its initial state $\pi_{\theta_0}$ to the current state $\pi_\theta$.*

**Proof Sketch for Theorem 4.2.**   The proof leverages a triangle inequality argument. For a prompt $x$, we first relate its online PVar, $\text{PVar}_\theta[x]$, to its offline PVar, $\widehat{\text{PVar}}_{\phi,\theta_0}[x]$, by introducing the PVar of the (unobserved) ground-truth reward $r^*$ as an intermediate anchor. The difference between any two PVar terms can be bounded by the maximum difference between their underlying reward functions (over responses $y$) and the total variation distance between their sampling distributions for that $x$. Summing these bounds yields a relationship of the form $\text{PVar}_\theta[x] \leq \widehat{\text{PVar}}_{\phi,\theta_0}[x] + \Xi(x; \theta, \phi)$. Substituting this into the bound from Theorem 4.1 yields the final result. The full proof is in Appendix B.

**Practical Implications and Discussion.**   Theorem 4.1 provides the core intuition: low PVar implies small gradients. Theorem 4.2 provides the formal justification for our practical data selection method. It shows that for any given prompt $x$, selecting it based on a high offline PVar ('$\widehat{\text{PVar}}_{\phi,\theta_0}[x]$') is effective because this term directly contributes to a higher upper bound on its own online training gradient. This justifies why filtering for high PVar prompts leads to more impactful training updates on average.

Our approach's effectiveness assumes the PVar signal is not dominated by error terms. The DPO objective inherently constrains policy drift through its implicit KL-divergence penalty, which, by Pinsker's inequality, also bounds the policy distribution shift. Our strong empirical results suggest these error terms are sufficiently controlled in practice, making offline PVar an effective proxy for identifying information-rich data.

## 5  EXPERIMENT

We design experiments to answer two questions: (1) *Does prompt-level preference variance correlate with gradient magnitudes in actual DPO training?* (2) *Does leveraging this variance lead to improved training outcomes or model performance?* To answer these questions, we conducted three sets of experiments. First, we analyzed the training loss by dividing the DPO training prompts into three distinct subsets based on their PVar (Top 50%, Bottom 50%, and Random 50%), and comparing convergence rates and final loss values. Then, we evaluated model performance across different benchmarks (AlpacaEval 2.0 and Arena-Hard) using the same subset division strategy. Finally, to simulate real-world deployment scenarios, we train model with the top 10% of human-annotated pairs from UltraFeedback (selected by PVar) and compare with the model training with the complete human-annotated dataset, demonstrating improvements in model performance with significantly fewer training examples. More experiments and analysis can be found at Appendix C.

### 5.1  SETUP

**Models:**   We use the following base models for fine-tuning LLMs: Mistral-7B-Instruct-v0.2, an instruction fine-tuned version of Mistral-7B-v0.2 (Jiang et al., 2023), and Llama-3.1-8B-Instruct, an instruction finetuned version of Llama-3.1-8B (Grattafiori et al., 2024). We use the Skywork-Reward-Llama-3.1-8B-v0.2 (Liu et al., 2024a) as the reward model to estimate the PVar except for those in Table 2. We use Llama-3.1-8B-Instruct in all of our experiments except for those in Table 1.

**Training Datasets:**   We use a diverse mix of datasets for training, including: **UltraFeedback** (Cui et al., 2023), a large-scale dataset with 60K diverse prompts; **Chatbot Arena Conversations** (Zheng et al., 2023), containing 33K real-world user conversations; **HH-RLHF** (Bai et al., 2022), a human preference dataset from Anthropic with over 160K comparisons focused on helpfulness and harmlessness; and **WebGPT** (Nakano et al., 2021), which consists of 20K question-answer pairs for fact-intensive, web-sourced tasks.

**Benchmarks:**   Following previous works (Meng et al., 2024; Deng et al., 2025; Wu et al., 2024; Chen et al., 2025), we use **AlpacaEval 2.0** (Dubois et al., 2024) and **Arena-Hard** (Li et al., 2024b;c) as our evaluation benchmarks. More information for these benchmarks can be found at Appendix C.

**Implementation Details:**   We employed the AdamW optimizer with standard parameters. Key hyperparameters for DPO training include a learning rate of $5 \times 10^{-7}$ with a cosine schedule and 0.1

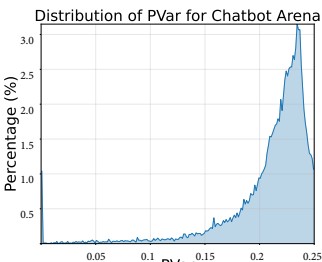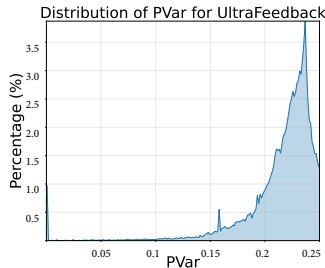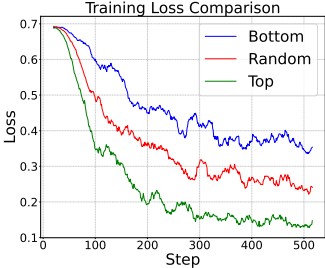

Figure 2: Left and middle: Distribution of Preference Variance (PVar) across prompts from two different datasets: Chatbot Arena Conversation and UltraFeedback. Each PVar is calculated using 5 responses generated by Llama 3.1-8B-Instruct. The distributions show a wide spread of PVar values, indicating significant variation in the informativeness of prompts, with a substantial portion exhibiting moderate-to-high PVar values. Right: Training loss curves for models fine-tuned on different data subsets selected based on PVar. The Top 50% subset (green) demonstrates faster convergence and reaches a lower final loss compared to Random 50% (red) and Bottom 50% (blue) selections, indicating more efficient learning from high-PVar training examples.

warmup ratio, a global batch size of 32, and a DPO $\beta$ of 0.1. All models were trained for two epochs. A comprehensive list of all hyperparameters and generation settings can be found in Appendix C.

## 5.2 MAIN RESULTS

**PVar Distribution Analysis.** We analyze the distribution of PVar across prompts in the left and middle panels of Figure 2. For each prompt in both datasets, we generate 5 responses and calculate the empirical PVar. The distributions from both datasets reveal similar patterns. We observe a wide range of PVar values (from near 0 to the maximum of 0.25), indicating significant variation in the strength of preference signals across prompts. Both distributions show that a substantial portion of prompts have relatively high PVar, suggesting that many examples in these datasets are informative for DPO. This consistency across different prompt collections indicates that our PVar-based analysis is robust.

**Training Loss Analysis.** The right panel Figure 2 illustrates the training loss curves for models fine-tuned on different subsets of data selected using PVar. We divided the training data into three distinct subsets based on their PVar: Top 50% (highest PVar), Bottom 50% (lowest PVar), and Random 50% (randomly selected examples), then separately trained models on each of these subsets. The visualization reveals distinct learning patterns. Models trained on the Top 50% subset demonstrate noticeably faster loss reduction and ultimately converge to a lower final loss value. In contrast, models trained on the Bottom 50% subset exhibit the slowest convergence rate and settle at a higher final loss. The Random 50% selection shows intermediate performance. These observations align with our theoretical analysis, providing evidence that prioritizing high-PVar examples leads to more efficient learning.

**Analysis on Different Benchmarks.** Following our previous methodology, we divide the data into three subsets: Top, Random, and Bottom based on PVar, each containing 50% of the prompts, and then separately trained models on each. We use responses with the highest and lowest rewards (given by Skywork-Reward-Llama-3.1-8B-v0.2) as the chosen and rejected responses, respectively. Table 1 presents a comprehensive evaluation across different base models and training datasets. The results demonstrate a consistent trend: training with top-PVar prompts yields performance that is consistently as good as or better than random selection, and often superior, particularly on the length-controlled win rate metric. Specifically, for Llama 3.1-8B-Instruct trained on UltraFeedback, the Top 50% selection achieves a 36.2% length-controlled win rate on AlpacaEval 2.0, outperforming both random (34.9%) and bottom (34.8%) selection. This trend holds across models, datasets, and evaluation metrics, providing compelling evidence that intelligent prompt selection can enhance model performance.

**PVar's Robustness: Outperforming the Reward Gap Across Varying Reward Model Quality.** To rigorously benchmark our method, we compare PVar against a strong and intuitive baseline: the **Reward Gap**. This strategy selects the top 50% of prompts exhibiting the largest difference between the maximum and minimum reward scores among generated responses (i.e., $\max_y r(x,y) -$

| Base Model | Training Set | Selection | AlpacaEval 2.0 | | Arena-Hard |
|---|---|---|---|---|---|
| | | | **LC (%)** | **WR (%)** | **WR (%)** |
| Llama 3.1-8B-Instruct | UltraFeedback | Top | **36.2** | **40.9** | **32.2** |
| | | Random | 34.9 | 39.3 | 31.0 |
| | | Bottom | 34.8 | 38.6 | 30.7 |
| | Chatbot Arena | Top | **36.2** | **39.6** | **30.0** |
| | | Random | 33.6 | 39.3 | 28.8 |
| | | Bottom | 32.9 | 37.5 | 29.2 |
| Mistral-7B-Instruct-v0.2 | UltraFeedback | Top | **31.2** | **36.1** | **19.6** |
| | | Random | 31.0 | **36.1** | 17.7 |
| | | Bottom | 30.1 | 35.1 | 18.8 |
| | Chatbot Arena | Top | **32.5** | **34.5** | **20.4** |
| | | Random | 29.1 | 31.6 | 18.0 |
| | | Bottom | 28.2 | 30.7 | 16.7 |

Table 1: Performance comparison of different prompt selection strategies. We partition each dataset into three segments (Top 50%, Random 50%, and Bottom 50%) based on PVar. For these experiments, winning and losing responses were determined by selecting generated responses with the highest and lowest scores from our reward model. Results show that training on top-ranked prompts consistently outperforms random and bottom selection across different base models and datasets. Bold numbers indicate the best performance within each model-dataset combination.

| Dataset | Reward Model | Selection Strategy | Win Rate (%) | LC Win Rate (%) |
|---|---|---|---|---|
| HH-RLHF | Intrinsic | PVar Top | **41.4** | **35.5** |
| | | Reward Gap Top | 39.3 | 35.1 |
| | Skywork-Llama-3.1-8B-v0.2 | PVar Top | **40.9** | **35.1** |
| | | Reward Gap Top | 38.9 | 34.7 |
| | | PVar Bottom | 38.0 | 35.0 |
| | Skywork-Llama-3.2-3B-v0.2 | PVar Top | **41.1** | **35.8** |
| | | Reward Gap Top | 39.1 | 33.7 |
| | | PVar Bottom | 39.0 | 34.9 |
| | Skywork-Llama-3.2-1B-v0.2 | PVar Top | **39.9** | **36.4** |
| | | Reward Gap Top | 38.9 | 35.3 |
| | | PVar Bottom | 38.2 | 33.8 |
| WebGPT | Intrinsic | PVar Top | **41.2** | **35.8** |
| | | Reward Gap Top | 38.9 | 34.2 |
| | Skywork-Llama-3.1-8B-v0.2 | PVar Top | **40.9** | **35.4** |
| | | Reward Gap Top | 39.5 | 34.9 |
| | | PVar Bottom | 37.8 | 33.7 |
| | Skywork-Llama-3.2-3B-v0.2 | PVar Top | **40.6** | **36.3** |
| | | Reward Gap Top | 38.9 | 34.6 |
| | | PVar Bottom | 38.2 | 33.6 |
| | Skywork-Llama-3.2-1B-v0.2 | PVar Top | **40.1** | **35.2** |
| | | Reward Gap Top | 38.6 | 34.3 |
| | | PVar Bottom | 38.1 | 34.1 |

Table 2: Robustness of selection strategies across different reward models. We train Llama-3.1-8B-Instruct on HH-RLHF and WebGPT datasets. 'PVar Top' consistently outperforms the 'Reward Gap Top' baseline, even when using the DPO-finetuned model's own implicit rewards (Intrinsic) for selection. Bold indicates the best result within each dataset-reward source group.

$\min_y r(x, y))$ Deng et al. (2025); Cui et al. (2025); Khaki et al. (2024). We evaluated both selection methods on HH-RLHF and WebGPT using three reward models of varying sizes (1B, 3B, and 8B). As detailed in Table 2, the PVar Top 50% subset yields superior performance in all configurations.

| Model Configuration | AlpacaEval 2.0 | | Arena-Hard |
|---|---|---|---|
| | LC (%) | WR (%) | WR (%) |
| Llama 3.1-8B-Instruct (Base Model) | 24.8 | 24.2 | 21.3 |
| + DPO w. 10% Human Data (Final) | **34.3** | **37.0** | **30.3** |
| + DPO w. 100% Human Data (Peak Perf. @ 64.4% Data) | 32.5 | 36.5 | 29.1 |
| + DPO w. 100% Human Data (Final Perf.) | 32.5 | 36.4 | 28.8 |

Table 3: Impact of selective DPO training using high-PVar prompts from UltraFeedback with human annotations. Results show that training with only the top 10% of prompts (selected via PVar) yields superior performance compared to using the full dataset, even when the full-dataset model has seen over six times more data. Bold indicates the best result.

The performance gap widens significantly with the less reliable 1B reward model, confirming that PVar is a more stable indicator of learning value than reward gap, particularly in the presence of noisy reward signals.

**Robustness with Intrinsic Reward Models.** To further assess robustness in resource-constrained scenarios, we conducted an experiment using an **Intrinsic Reward Model**. Since the implicit reward of the reference model itself is zero, we first fine-tuned the base model on a randomly selected subset (5%) of the data via DPO. We then utilized this preliminary policy's implicit reward formulation, $\hat{r}_\theta(x, y) = \beta(\log \pi_\theta(y|x) - \log \pi_{\text{ref}}(y|x))$, to calculate both PVar and Reward Gap for the remaining data. Notably, as shown in the first row of Table 2, PVar-based selection significantly outperforms Reward Gap selection even in this intrinsic setting. This result suggests that PVar captures the distributional disagreement more effectively than linear reward gaps, making it highly robust even when the supervision signal is weak or self-derived.

**Efficient DPO with Selected Human Annotations.** To simulate practical deployment scenarios, this experiment uses the UltraFeedback dataset with its original human-annotated response pairs. We first identified the top 10% of prompts with the highest PVar, then trained a model using only the human-labeled response pairs corresponding to these prompts. We compared this selective approach to training with all human-labeled pairs, evaluating the full-data model at checkpoints every 2000 steps. We trained both models for two full epochs over their respective datasets.

Table 3 shows a compelling result: training with just 10% of the data achieves a final win rate of 37.0% on AlpacaEval 2.0, which is significantly higher than the *peak performance* (36.5%) of the model trained on the full dataset. The full-data model reaches this peak after training on approximately 64.4% of the data, meaning our method achieves a better final model with over six times less data. This supports our hypothesis that focusing on high-signal, high-PVar data leads to both a more efficient process and a better final model.

## 6 CONCLUSION

In this paper, we establish a direct link between Preference Variance (PVar) and the effectiveness of DPO training. Our theoretical analysis shows, and empirical results confirm, that high-PVar prompts generate larger gradient updates, leading to faster convergence and superior model performance. Across multiple models and datasets, training on high-PVar subsets consistently outperformed using random or low-PVar data. Most notably, we found that training on only the top 10% of human-annotated high-PVar prompts surpassed training on the full dataset, demonstrating a path to significantly reduce annotation costs without sacrificing performance. This work provides a practical, theoretically grounded method for identifying high-value training data, enabling more efficient resource allocation in LLM alignment.

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

# A PROOF OF THEOREM 4.1

**Additional Notations.** Let $f(\cdot; \theta)$ denote the neural network that produces logits. For a response $y$ of length $|y|$, $y_i$ is the $i$-th token and $y_{<i}$ are the preceding tokens. We use $\mathbf{e}_{y_i} \in \mathbb{R}^{|\mathcal{V}|}$ for the one-hot vector of token $y_i$ from vocabulary $\mathcal{V}$, and $J_{f(x,y_{<i};\theta)}$ for the Jacobian of $f(x, y_{<i}; \theta)$ with respect to $\theta$. We assume the logits mapping is Lipschitz continuous, which implies the Jacobian norm is bounded.

**Theorem A.1** (Formal version of Theorem 4.1). *For parameters $\theta \in \mathbb{R}^p$ and input $x \in \mathcal{X}$, the norm of the DPO loss gradient is upper bounded by:*

$$\|\nabla_\theta \mathcal{L}_{\mathrm{DPO}}(\pi_\theta, \pi_{\mathrm{ref}}; x)\| \leq C(x, \theta) \cdot PVar_\theta[x]^{1/3},$$

*where $PVar_\theta[x] = \mathrm{Var}_{y_i, y_j \sim \pi_\theta(\cdot|x)}[p_\theta(x; y_i, y_j)]$ and $C(x, \theta)$ is a constant defined as $C(x, \theta) := 8\beta|y|\gamma(x; \theta)$, with $|y|$ denoting the maximum response length $L_{\max}$ and $\gamma(x; \theta) := \max_{i, y_{<i}} \|J_{f(x,y_{<i};\theta)}\|_2$.*

*Proof.* The DPO loss gradient for a given prompt $x$ can be expressed as:

$$\nabla_\theta \mathcal{L}_{\mathrm{DPO}}(\pi_\theta, \pi_{\mathrm{ref}}; x) = -\beta \mathbb{E}_{y_w, y_l \sim \pi_\theta(\cdot|x)}[(1 - p_\theta(x; y_w, y_l)) \cdot [\nabla_\theta \log \pi_\theta(y_w|x) - \nabla_\theta \log \pi_\theta(y_l|x)]]$$

where we used the fact that the expectation is over the model's own distribution $\pi_\theta$ and $\sigma(-u) = 1 - \sigma(u)$.

For $c > 0$ to be determined later, we define $\mathcal{Y}_c := \{(y_w, y_l) : |p_\theta(x; y_w, y_l) - \frac{1}{2}| > c\}$, and a modified preference function:

$$\tilde{p}_\theta(x; y_w, y_l) := \begin{cases} p_\theta(x; y_w, y_l) & \text{if } (y_w, y_l) \notin \mathcal{Y}_c \\ \frac{1}{2} & \text{if } (y_w, y_l) \in \mathcal{Y}_c \end{cases}$$

We decompose the gradient into two terms, $A$ and $B$, based on $\tilde{p}_\theta$ and $(p_\theta - \tilde{p}_\theta)$:

$$\nabla_\theta \mathcal{L}_{\mathrm{DPO}} = \underbrace{-\beta \mathbb{E}[(1 - \tilde{p}_\theta) \cdot \Delta \nabla \log \pi_\theta]}_{A} \underbrace{-\beta \mathbb{E}[(p_\theta - \tilde{p}_\theta) \cdot \Delta \nabla \log \pi_\theta]}_{B}$$

where $\Delta \nabla \log \pi_\theta = \nabla_\theta \log \pi_\theta(y_w|x) - \nabla_\theta \log \pi_\theta(y_l|x)$.

By construction, $|\tilde{p}_\theta(x; \cdot, \cdot) - 1/2| \leq c$, so $|1 - \tilde{p}_\theta|$ is also close to $1/2$. Following a detailed derivation similar to that in (Razin et al., 2023), we can bound the term $A$ by leveraging the Lipschitz properties of the softmax function and the boundedness of $1 - \tilde{p}_\theta$. This yields:

$$\|A\| \leq 4\beta c|y|\gamma(x; \theta).$$

For term $B$, it is non-zero only on the set $\mathcal{Y}_c$. By Chebyshev's inequality, the probability mass of this set is bounded by PVar:

$$\mathbb{P}((y_w, y_l) \in \mathcal{Y}_c|x) = \mathbb{P}\left(\left|p_\theta - \frac{1}{2}\right| > c\right) \leq \frac{\mathrm{Var}[p_\theta]}{c^2} = \frac{PVar_\theta[x]}{c^2}.$$

The gradient term $\|\Delta \nabla \log \pi_\theta\|$ can be bounded by $4|y|\gamma(x; \theta)$. Since $|p_\theta - \tilde{p}_\theta| \leq 1$, we have:

$$\begin{aligned} \|B\| &\leq \beta \cdot \mathbb{E}[\mathbb{I}((y_w, y_l) \in \mathcal{Y}_c) \cdot |p_\theta - \tilde{p}_\theta| \cdot \|\Delta \nabla \log \pi_\theta\|] \\ &\leq \beta \cdot (4|y|\gamma(x; \theta)) \cdot \mathbb{P}((y_w, y_l) \in \mathcal{Y}_c|x) \\ &\leq 4\beta|y|\gamma(x; \theta) \frac{PVar_\theta[x]}{c^2}. \end{aligned}$$

Combining the bounds for $A$ and $B$:

$$\|\nabla_\theta \mathcal{L}_{\mathrm{DPO}}(\pi_\theta, \pi_{\mathrm{ref}}; x)\| \leq 4\beta c|y|\gamma(x; \theta) + 4\beta|y|\gamma(x; \theta) \frac{PVar_\theta[x]}{c^2}.$$

This bound is minimized by choosing $c = (2 \cdot PVar_\theta[x])^{1/3}$, which yields:

$$\|\nabla_\theta \mathcal{L}_{\mathrm{DPO}}(\pi_\theta, \pi_{\mathrm{ref}}; x)\| \leq (4 \cdot 2^{1/3} + 4 \cdot 2^{-2/3})\beta|y|\gamma(x; \theta) \cdot PVar_\theta[x]^{1/3}.$$

Since $(4 \cdot 2^{1/3} + 4 \cdot 2^{-2/3}) \approx 7.56 < 8$, we can use a simpler constant, yielding the final bound. $\square$

## B PROOF OF THEOREM 4.2

We first introduce three foundational lemmas. Let $\mu_{\theta,x} := \pi_\theta(\cdot|x) \otimes \pi_\theta(\cdot|x)$.

**Lemma B.1** (Variance Difference by Measure Change). *For any bounded function $U : \mathcal{Y}^2 \to [0, 1]$ and probability measures $\mu, \nu$ on $\mathcal{Y}^2$, we have $|\operatorname{Var}_\mu(U) - \operatorname{Var}_\nu(U)| \leq 6 \cdot \operatorname{TV}(\mu, \nu)$, where TV is the total variation distance.*

*Proof.* $|\mathbb{E}_\mu[U^2] - \mathbb{E}_\nu[U^2]| \leq 2\operatorname{TV}(\mu, \nu)\|U^2\|_\infty \leq 2\operatorname{TV}$. Similarly, $|\mathbb{E}_\mu[U] - \mathbb{E}_\nu[U]| \leq 2\operatorname{TV}$. Since $|(\mathbb{E}_\mu U)^2 - (\mathbb{E}_\nu U)^2| = |\mathbb{E}_\mu U - \mathbb{E}_\nu U||\mathbb{E}_\mu U + \mathbb{E}_\nu U| \leq 2\operatorname{TV} \cdot 2 = 4\operatorname{TV}$. Combining gives $6\operatorname{TV}$. $\square$

**Lemma B.2** (PVar Difference by Reward Change). *Let $p_1 = \sigma(r_1(y_i) - r_1(y_j))$ and $p_2 = \sigma(r_2(y_i) - r_2(y_j))$. If $\sup_y |r_1(x, y) - r_2(x, y)| \leq \Delta$ for a fixed $x$, then for a fixed measure $\mu$, $|\operatorname{PVar}[x; r_1, \mu] - \operatorname{PVar}[x; r_2, \mu]| \leq 2\Delta$.*

*Proof.* By the mean value theorem, $|\sigma(u) - \sigma(v)| \leq \frac{1}{4}|u - v|$ since $\sup_t \sigma'(t) = 1/4$. Thus, $|p_1 - p_2| \leq \frac{1}{4}|(r_1(x, y_i) - r_2(x, y_i)) - (r_1(x, y_j) - r_2(x, y_j))| \leq \frac{1}{4}(2\Delta) = \frac{\Delta}{2}$. The rest of the proof follows the logic of Lemma B.1, with $|\mathbb{E}[p_1^2] - \mathbb{E}[p_2^2]| \leq \mathbb{E}[|p_1 - p_2||p_1 + p_2|] \leq 2\mathbb{E}[|p_1 - p_2|] \leq \Delta$, and $|(\mathbb{E}p_1)^2 - (\mathbb{E}p_2)^2| \leq \Delta$. Summing them yields $2\Delta$. $\square$

**Lemma B.3** (TV distance of Product Measures). *For a fixed prompt $x$, $\operatorname{TV}(\pi_1(\cdot|x) \otimes \pi_1(\cdot|x), \pi_2(\cdot|x) \otimes \pi_2(\cdot|x)) \leq 2 \cdot \operatorname{TV}(\pi_1(\cdot|x), \pi_2(\cdot|x))$.*

*Proof.* Let $\mu_i = \pi_i(\cdot|x) \otimes \pi_i(\cdot|x)$. The TV distance is $\frac{1}{2} \int |d\mu_1 - d\mu_2|$.

$$\operatorname{TV}(\mu_1, \mu_2) = \frac{1}{2} \iint |\pi_1(y_1|x)\pi_1(y_2|x) - \pi_2(y_1|x)\pi_2(y_2|x)|dy_1 dy_2$$

$$= \frac{1}{2} \iint |\pi_1(y_1|x)\pi_1(y_2|x) - \pi_1(y_1|x)\pi_2(y_2|x) + \pi_1(y_1|x)\pi_2(y_2|x) - \pi_2(y_1|x)\pi_2(y_2|x)|dy_1 dy_2$$

$$\leq \frac{1}{2} \iint |\pi_1(y_1|x)(\pi_1(y_2|x) - \pi_2(y_2|x))|dy_1 dy_2 + \frac{1}{2} \iint |\pi_2(y_2|x)(\pi_1(y_1|x) - \pi_2(y_1|x))|dy_1 dy_2$$

$$= \frac{1}{2} \int \pi_1(y_1|x)dy_1 \int |\pi_1(y_2|x) - \pi_2(y_2|x)|dy_2 + \frac{1}{2} \int \pi_2(y_2|x)dy_2 \int |\pi_1(y_1|x) - \pi_2(y_1|x)|dy_1$$

$$= \operatorname{TV}(\pi_1(\cdot|x), \pi_2(\cdot|x)) + \operatorname{TV}(\pi_1(\cdot|x), \pi_2(\cdot|x)) = 2 \cdot \operatorname{TV}(\pi_1(\cdot|x), \pi_2(\cdot|x)).$$

$\square$

*Proof of Theorem 4.2.* Let $e_\theta(y) = \hat{r}_\theta(x, y)$. For a fixed prompt $x$, we use the (unobserved) ground-truth reward $r^*(x, y)$ as an anchor and apply a triangle inequality:

$$\begin{aligned}
|\operatorname{PVar}_\theta[x] - \widehat{\operatorname{PVar}}_{\phi,\theta_0}[x]| &= |\operatorname{PVar}[x; e_\theta, \mu_{\theta,x}] - \operatorname{PVar}[x; r_\phi, \mu_{\theta_0,x}]| \\
&\leq |\operatorname{PVar}[x; e_\theta, \mu_{\theta,x}] - \operatorname{PVar}[x; r^*, \mu_{\theta,x}]| &\text{(Term 1)} \\
&\quad + |\operatorname{PVar}[x; r^*, \mu_{\theta,x}] - \operatorname{PVar}[x; r^*, \mu_{\theta_0,x}]| &\text{(Term 2)} \\
&\quad + |\operatorname{PVar}[x; r^*, \mu_{\theta_0,x}] - \operatorname{PVar}[x; r_\phi, \mu_{\theta_0,x}]| &\text{(Term 3)}
\end{aligned}$$

We bound each term for the specific $x$:

- **Term 1 (Reward change from $r^*$ to $e_\theta$):** By Lemma B.2, this is bounded by $2 \sup_y |e_\theta(x, y) - r^*(x, y)|$.

- **Term 2 (Measure change from $\mu_{\theta_0,x}$ to $\mu_{\theta,x}$):** By Lemma B.1, this is bounded by $6 \cdot \operatorname{TV}(\mu_{\theta,x}, \mu_{\theta_0,x})$.

- **Term 3 (Reward change from $r_\phi$ to $r^*$):** By Lemma B.2, this is bounded by $2 \sup_y |r_\phi(x, y) - r^*(x, y)|$.

Summing these bounds, we get:

$$\mathrm{PVar}_\theta[x] \le \widehat{\mathrm{PVar}}_{\phi,\theta_0}[x] + 2\sup_y |e_\theta(x,y) - r^*(x,y)| + 6\mathrm{TV}(\mu_{\theta,x}, \mu_{\theta_0,x}) + 2\sup_y |r_\phi(x,y) - r^*(x,y)|.$$

The term $\sup_y |e_\theta(x,y) - r^*(x,y)|$ can be further bounded by $\sup_y |e_\theta(x,y) - r_\phi(x,y)| + \sup_y |r_\phi(x,y) - r^*(x,y)|$ using triangle inequality. This leads to the prompt-specific error term $\Xi(x; \theta, \phi)$ defined in the theorem statement. Now, substitute this into the result of Theorem 4.1:

$$\|\nabla_\theta \mathcal{L}_{\mathrm{DPO}}(\pi_\theta, \pi_{\mathrm{ref}}; x)\| \le C(x, \theta) \cdot \left(\widehat{\mathrm{PVar}}_{\phi,\theta_0}[x] + \Xi(x; \theta, \phi)\right)^{1/3}.$$

This concludes the proof, as the bound holds for the specific prompt $x$. □

## C  ADDITIONAL EXPERIMENTAL RESULTS

**Training Configuration.**  All experiments were conducted on 4 NVIDIA H100 GPUs. For DPO training, we used the following settings: a per-device batch size of 2 with 4 gradient accumulation steps (total global batch size of 32), a maximum prompt length of 4096, and 2 training epochs. We note that for all evaluation metrics, the standard error typically ranges between 1% and 2%.

**More Information about AlpacaEval 2.0 and Arena-Hard.**  AlpacaEval 2.0 comprises 805 diverse prompts from AlpacaFarm (Dubois et al., 2023), covering general human instructions across various scenarios. It employs GPT-4-Turbo as a proxy for human judgment. Performance is primarily measured through win rates (WR) against the reference responses, with length-controlled win rates (LC) specifically designed to mitigate bias toward lengthy responses and encourage concise, effective answers. Arena-Hard consists of particularly difficult prompts that require advanced reasoning, knowledge application, and instruction following. Models are compared against GPT-4-0314 as the baseline. Performance is measured through win rates (WR) as determined by the GPT-4-Turbo judge.

**PVar Estimation Generation.**  To estimate PVar for each prompt, we generated 5 responses using a maximum length of 2048, a temperature of 0.7, and top-p sampling with p=1.

**Evaluation Generation.**  For evaluations on AlpacaEval 2.0 and Arena-Hard, we used a temperature of 0.7 and top-p sampling with p=1. We adhered to the standard maximum length settings for each benchmark: 2048 for AlpacaEval 2.0 and 4096 for Arena-Hard.

**Specifics for Figures.**  The models used in both Figure 3 and the right panel of Figure 2 are Llama-3.1-8B Instruct, and the training dataset employed for both analyses is the Chatbot Arena Conversation dataset.

**Training Margin Analysis.**  Figure 3 presents the training margin evolution for models fine-tuned on different subsets of data selected using PVar. The margin, calculated as the difference in model confidence scores between preferred and rejected responses, serves as an indicator of how well the model learns to distinguish between response qualities. The results reveal that models trained on the Top 50% subset (highest PVar) exhibit the most rapid margin increase during the early training phases and ultimately achieve the highest final margin values. Conversely, models trained on the Bottom 50% subset show the slowest margin growth and converge to lower final margins. The Random 50% selection demonstrates performance that falls between these two extremes. These margin dynamics support our theory that high-PVar examples provide clearer preference signals, enabling models to develop more robust preference understanding.

**Ablation Study on Hyperparameter.**  To validate the robustness of our findings, we conduct an ablation study by varying the beta parameter to $\beta = 0.01$ in DPO training following (Deng et al., 2025). Table 4 presents the results of this ablation experiment, maintaining the same experimental setup where data is divided into Top, Random, and Bottom subsets based on PVar. The results demonstrate that even with a different beta value, the Top selection strategy consistently outperforms Random and Bottom selections across all model-dataset combinations. This consistency across different beta values reinforces that the benefits of prioritizing high-PVar prompts in the DPO training process.

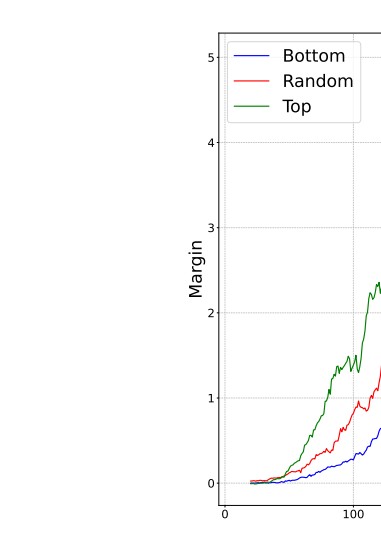

Figure 3: Training margin curves for models fine-tuned on different data subsets selected based on PVar. The margin represents the difference in model confidence between preferred and rejected responses during training. Models trained on the Top 50% subset (green) show the fastest margin increase and achieve the highest final margin values, while the Bottom 50% subset (blue) exhibits slower margin growth and lower final margins. The Random 50% selection (red) demonstrates intermediate performance, confirming that high-PVar examples facilitate more effective preference learning.

Table 4: Ablation study on beta parameter ($\beta = 0.01$) for different prompt selection strategies. Training on top-PVar prompts consistently achieves the best performance across different models and datasets. Bold numbers indicate the best performance within each model-dataset combination.

| Base Model | Training Set | Selection | AlpacaEval 2.0 | | Arena-Hard |
| --- | --- | --- | --- | --- | --- |
| | | | LC (%) | WR (%) | WR (%) |
| Llama 3.1-8B-Instruct | UltraFeedback | Top | **36.8** | **40.8** | **33.3** |
| | | Random | 34.0 | 38.4 | 29.4 |
| | | Bottom | 34.9 | 38.4 | 28.6 |
| | Chatbot Arena | Top | **36.1** | **39.5** | **32.4** |
| | | Random | 35.2 | 39.4 | 30.6 |
| | | Bottom | 35.0 | 38.6 | 29.4 |
| Mistral-7B-Instruct-v0.2 | UltraFeedback | Top | **30.1** | **37.9** | **20.9** |
| | | Random | 29.0 | 35.4 | 18.9 |
| | | Bottom | 29.0 | 36.8 | 18.9 |
| | Chatbot Arena | Top | **31.3** | **34.4** | **20.4** |
| | | Random | 29.8 | 34.3 | 18.0 |
| | | Bottom | 30.3 | **34.4** | 16.7 |

# D QUALITATIVE ANALYSIS

As qualitative examples truly illuminate the practical impact of our method. We compared models trained on Top vs. Bottom PVar data and observed clear behavioral differences.

Training on high-PVar prompts leads to a model that is more **nuanced and capable** in several key areas:

1. **Critical Thinking & Fact-Checking**: When asked "Do you know why turkeys become the official food of thanksgiving?", the **Low-PVar** model gives a lengthy explanation

accepting the false premise. In contrast, the **High-PVar** model **immediately corrects the misconception**, stating, "A common myth has been debunked! Turkeys are not actually the official food of Thanksgiving..." before providing a more critical analysis.

2. **Information Organization**: For knowledge queries, the **Low-PVar** model often produces long, unstructured lists. The **High-PVar** model **organizes information with clear categorical headings** (e.g., **Jazz and Blues**, **Swing and Dance Music**), making the response more reader-friendly and logical.

These examples suggest that high-PVar prompts, which often reflect human disagreement on complex criteria, train the model to better handle the more sophisticated and nuanced aspects of language and reasoning.

# E   A UNIFIED THEORY OF PVAR: CONNECTING GRADIENT MAGNITUDE, DIRECTION, AND SNR

Many preference-learning objectives admit the same pairwise gradient structure. We show that the *preference variance* (PVar)—the variance of the model's predicted preference probability for response pairs— controls (i) a sharp upper bound on the gradient norm, (ii) a positive *directional* descent component (after trimming), and (iii) a per-prompt *signal-to-noise ratio* (SNR) lower bound.

**Generic pairwise gradient form.**   Let $\pi_\theta(y|x)$ be the policy, $\beta > 0$ a scalar, and sample an *ordered* pair $(y_i, y_j) \sim \pi_\theta(\cdot|x)^{\otimes 2}$. Define the *relative score*

$$s_\theta(x; y_i, y_j) := \beta\big(\log \pi_\theta(y_i|x) - \log \pi_\theta(y_j|x)\big), \qquad p_\theta(x; y_i, y_j) := \sigma\big(s_\theta(x; y_i, y_j)\big),$$

and the *score vector*

$$\phi(x; y_i, y_j; \theta) := \beta\Big(\nabla_\theta \log \pi_\theta(y_i|x) - \nabla_\theta \log \pi_\theta(y_j|x)\Big).$$

For a broad class of logistic-type pairwise losses $\ell(s, Z)$ with $Z \in \{0, 1\}$, the per-prompt gradient takes the form

$$\nabla_\theta \mathcal{L}(x) = \mathbb{E}\big[w(p_\theta, Z)\, \phi(x; y_i, y_j; \theta)\big], \qquad w(p, Z) := \frac{\partial \ell}{\partial s}(s, Z) \ \ \text{with} \ \ p = \sigma(s). \quad (4)$$

*All expectations in this section are taken w.r.t. the **online** sampling distribution* $(y_i, y_j) \sim \pi_\theta(\cdot|x)^{\otimes 2}$ unless otherwise stated. By ordered-pair symmetry, $\mathbb{E}[p_\theta(x; y_i, y_j)] = \frac{1}{2}$ because $p_\theta(x; y_i, y_j) + p_\theta(x; y_j, y_i) = 1$.

**A uniform score bound.**   Assume responses are truncated to length $L_{\max}$. Let $\gamma(x; \theta)$ be an upper bound on the Jacobian norm of the model logits w.r.t. $\theta$ (cf. Appendix A). By standard softmax–Jacobian bounds and the score-function identity $\mathbb{E}_{y \sim \pi_\theta}[\nabla_\theta \log \pi_\theta(y|x)] = 0$,

$$\big\|\nabla_\theta \log \pi_\theta(y|x)\big\|_2 \leq 2 L_{\max} \gamma(x; \theta) \implies \|\phi(x; y_i, y_j; \theta)\|_2 \leq 4\beta L_{\max} \gamma(x; \theta). \quad (5)$$

**Preference variance and ring-trimmed second moment.**   Define the per-prompt PVar

$$\mathrm{PVar}_\theta[x] := \mathrm{Var}_{(y_i, y_j) \sim \pi_\theta^{\otimes 2}}\big(p_\theta(x; y_i, y_j)\big).$$

For trimming radii $\varepsilon \in (0, \frac{1}{2})$ and $\delta_{\mathrm{ring}} \in (0, \frac{1}{2})$, define the *ring region*

$$\mathcal{R}_{\varepsilon, \delta_{\mathrm{ring}}} := \Big\{ \varepsilon \leq \big|p_\theta(x; y_i, y_j) - \tfrac{1}{2}\big| \leq \tfrac{1}{2} - \delta_{\mathrm{ring}} \Big\},$$

and its ring-trimmed second moment

$$\mathrm{M2}_{\varepsilon, \delta_{\mathrm{ring}}}[x] := \mathbb{E}\Big[\big(p_\theta(x; y_i, y_j) - \tfrac{1}{2}\big)^2 \mathbf{1}_{\mathcal{R}_{\varepsilon, \delta_{\mathrm{ring}}}}\Big].$$

CORE ASSUMPTIONS

**Assumption E.1** (Label model and conditional independence). There exists a differentiable calibration map $\rho : [0, 1] \to [0, 1]$ with $\rho(\frac{1}{2}) = \frac{1}{2}$ such that

$$Z \,|\, p_\theta \;\sim\; \mathrm{Bernoulli}\big(\rho(p_\theta)\big), \qquad Z \perp\!\!\!\perp \phi \,|\, p_\theta.$$

Let $\bar{w}(p) := \mathbb{E}\big[w(p, Z) \mid p\big]$ and $\Delta_w(p) := w(p, 1) - w(p, 0)$.

**Remark (weaker alternative).** It suffices to assume

$$\mathbb{E}\big[(w - \bar{w}(p_\theta))\, \phi \mid p_\theta\big] = 0,$$

which already implies $\mathbb{E}[w\phi] = \mathbb{E}[\bar{w}(p_\theta)\phi]$.

**Assumption E.2** (Generic weight regularity). There exist finite constants $W, L > 0$ such that:

   (G1) (*Boundedness*) $|w(p, Z)| \le W$ for all $(p, Z)$;

   (G2) (*Lipschitz conditional mean at center*) $|\bar{w}(p) - \bar{w}(\frac{1}{2})| \le L\,|p - \frac{1}{2}|$ for $p \in [0, 1]$.

**Assumption E.3** (Opposite-slope margin on a ring). Fix $\varepsilon, \delta_{\mathrm{ring}} \in (0, \frac{1}{2})$ and the ring $\mathcal{R}_{\varepsilon, \delta_{\mathrm{ring}}}$. There exists $\tilde{\delta}_{\varepsilon, \delta_{\mathrm{ring}}} > 0$ such that for all $p$ with $\varepsilon \le |p - \frac{1}{2}| \le \frac{1}{2} - \delta_{\mathrm{ring}}$,

$$-\bar{w}(p)\,\mathrm{sign}\big(p - \tfrac{1}{2}\big) \;\ge\; \tilde{\delta}_{\varepsilon, \delta_{\mathrm{ring}}}\,|p - \tfrac{1}{2}|.$$

**Remark (a sufficient condition).** In the logistic cross-entropy case with $w(p, Z) = p - Z$ (so $\bar{w}(p) = p - \rho(p)$), Assumption E.3 holds if the teacher calibration is *sharper than the identity* on the ring, e.g., if

$$\mathrm{sign}(p - \tfrac{1}{2})\big(\rho(p) - \tfrac{1}{2}\big) \;\ge\; (1 + \tilde{\delta}_{\varepsilon, \delta_{\mathrm{ring}}})\,|p - \tfrac{1}{2}| \quad \text{for } p \in \mathcal{R}_{\varepsilon, \delta_{\mathrm{ring}}}.$$

Locally, a sufficient condition is $\rho$ differentiable near $\frac{1}{2}$ with $\rho'(1/2) > 1$.

**Assumption E.4** (Directional excitability (non-degeneracy)). Define the (ring-trimmed) signed mean-score direction

$$u_x \;:=\; \frac{\mathbb{E}\big[\phi\,\mathrm{sign}(p_\theta - \frac{1}{2})\,\mathbf{1}_{\mathcal{R}_{\varepsilon, \delta_{\mathrm{ring}}}}\big]}{\big\|\mathbb{E}\big[\phi\,\mathrm{sign}(p_\theta - \frac{1}{2})\,\mathbf{1}_{\mathcal{R}_{\varepsilon, \delta_{\mathrm{ring}}}}\big]\big\|_2} \;\in\; \mathbb{R}^p.$$

(If the denominator vanishes, set $u_x$ arbitrarily and $\kappa_{\varepsilon, \delta_{\mathrm{ring}}} = 0$.) There exists $\kappa_{\varepsilon, \delta_{\mathrm{ring}}} \ge 0$ such that

$$\mathbb{E}\Big[\langle \phi, u_x \rangle\,\mathrm{sign}(p_\theta - \tfrac{1}{2})\,|p_\theta - \tfrac{1}{2}|\,\mathbf{1}_{\mathcal{R}_{\varepsilon, \delta_{\mathrm{ring}}}}\Big] \;\ge\; \kappa_{\varepsilon, \delta_{\mathrm{ring}}}\,\mathrm{M2}_{\varepsilon, \delta_{\mathrm{ring}}}[x].$$

TWO GENERIC PRINCIPLES

**Proposition E.5** (Principle 1: Low PVar implies small updates). *Under Assumptions E.1–E.2 and the score bound equation 5, for any prompt $x$,*

$$\big\|\nabla_\theta \mathcal{L}(x)\big\| \;\le\; 8\,\beta\,L_{\max}\,\gamma(x; \theta)\,\Big(L^{2/3}W^{1/3}\Big)\,\mathrm{PVar}_\theta[x]^{1/3}.$$

**Proposition E.6** (Principle 2: Directional descent and SNR with trimming). *Under Assumptions E.1, E.2, E.3, E.4 and the bound equation 5, let $W := \sup_{p, Z} |w(p, Z)|$ and $\Delta_\star := \sup_{p \in [0, 1]} |w(p, 1) -$*

$w(p, 0)|$. *Then, for any prompt $x$,*

$$\big\langle -\nabla_\theta \mathcal{L}(x), u_x \big\rangle \ \geq \ \tilde{\delta}_{\varepsilon,\delta_{\rm ring}} \, \kappa_{\varepsilon,\delta_{\rm ring}} \, {\rm M2}_{\varepsilon,\delta_{\rm ring}}[x]$$

$$- \ 4\,\beta\,L_{\max}\,\gamma(x;\theta) \Big( \underbrace{L\,\varepsilon}_{\textit{near-center slack}} + \underbrace{W \, \frac{{\rm PVar}_\theta[x]}{(1/2 - \delta_{\rm ring})^2}}_{\textit{endpoint slack}} \Big),$$

$$
\begin{aligned}
{\rm SNR}(x) \ &:= \ \frac{\big|\langle \mathbb{E}[w\phi], u_x \rangle\big|}{\sqrt{{\rm Var}(\langle w\phi, u_x\rangle)}} \\[2mm]
&\geq \ \frac{\tilde{\delta}_{\varepsilon,\delta_{\rm ring}} \, \kappa_{\varepsilon,\delta_{\rm ring}} \, {\rm M2}_{\varepsilon,\delta_{\rm ring}}[x] \ - \ 4\,\beta\,L_{\max}\,\gamma \big( L\,\varepsilon + W\,\frac{{\rm PVar}_\theta[x]}{(1/2-\delta_{\rm ring})^2} \big)}{4\,\beta\,L_{\max}\,\gamma(x;\theta)\,\sqrt{W^2 + \tfrac{1}{4}\Delta_\star^2}} \\[2mm]
&\geq \ \frac{\tilde{\delta}_{\varepsilon,\delta_{\rm ring}} \, \kappa_{\varepsilon,\delta_{\rm ring}} \, {\rm M2}_{\varepsilon,\delta_{\rm ring}}[x] \ - \ 4\,\beta\,L_{\max}\,\gamma \big( L\,\varepsilon + W\,\frac{{\rm PVar}_\theta[x]}{(1/2-\delta_{\rm ring})^2} \big)}{4\sqrt{2}\,W\,\beta\,L_{\max}\,\gamma(x;\theta)} .
\end{aligned}
$$

*The first denominator bound uses the law of total variance with the two components (conditional variance and mean-variance) both controlled; the second bound uses $\Delta_\star \leq 2W$.*

**Remark.** Choosing $(\varepsilon, \delta_{\rm ring})$ so that the slack term $4\beta L_{\max}\gamma\big(L\,\varepsilon + W\,{\rm PVar}_\theta[x]/(1/2 - \delta_{\rm ring})^2\big)$ is dominated by $\tilde{\delta}_{\varepsilon,\delta_{\rm ring}}\kappa_{\varepsilon,\delta_{\rm ring}}\,{\rm M2}_{\varepsilon,\delta_{\rm ring}}[x]$ yields a strictly positive directional component.

**Remark (DPO-CE constants and reference shift).** For the logistic cross-entropy with $w(p, Z) = p - Z$, we have $W = 1$ and $\Delta_\star = 1$. The constant $L$ in (G2) is the *local* Lipschitz constant of $\bar{w}(p) = p - \rho(p)$ around $p = \frac{1}{2}$. If $\rho$ is differentiable on a neighborhood $\mathcal{N}$ of $\frac{1}{2}$, then $L = \sup_{p \in \mathcal{N}} |1 - \rho'(p)|$; if $\rho$ is only Lipschitz on $\mathcal{N}$ with constant ${\rm Lip}(\rho)$, then $L \leq 1 + {\rm Lip}(\rho)$. In DPO, one may also use the reference-shifted score

$$s_\theta^{\rm ref}(x; y_i, y_j) \ := \ \beta\Big( \log \tfrac{\pi_\theta(y_i|x)}{\pi_{\rm ref}(y_i|x)} - \log \tfrac{\pi_\theta(y_j|x)}{\pi_{\rm ref}(y_j|x)} \Big), \quad p_\theta^{\rm ref} = \sigma(s_\theta^{\rm ref}).$$

Since $\pi_{\rm ref}$ is independent of $\theta$, the score vector $\phi$ is unchanged, while $w$ only changes through replacing $p$ by $p^{\rm ref}$. All results above continue to hold with this substitution.

**Scope and connection to offline selection.** The bounds above control the *online* gradient taken under $\pi_\theta^{\otimes 2}$. For *offline* data selection, combine them with the offline-to-online bridge in Theorem 4.2 from the main text, which replaces the online PVar by its offline estimate plus controlled error terms.

# F  PROOF SKETCHES FOR SECTION E

**Tools.** We repeatedly use: (i) Chebyshev's inequality $\mathbb{P}(|p_\theta - \frac{1}{2}| \geq c) \leq {\rm PVar}_\theta[x]/c^2$; (ii) the bound equation 5; (iii) ordered-pair symmetry $\mathbb{E}[p_\theta] = \frac{1}{2}$; and (iv) the law of total variance.

**Proof sketch of Proposition E.5.** By Assumption E.1, $\mathbb{E}[w\phi] = \mathbb{E}[\bar{w}(p_\theta)\phi]$ since $\mathbb{E}[(w - \bar{w}(p_\theta))\phi \mid p_\theta] = 0$. Subtract and add $\bar{w}(\frac{1}{2})$ to write $\mathbb{E}[(\bar{w}(p_\theta) - \bar{w}(\frac{1}{2}))\phi]$. For any $c > 0$, split the domain into $\{|p_\theta - \frac{1}{2}| \leq c\}$ and its complement. On the near region, (G2) gives $|\bar{w}(p_\theta) - \bar{w}(\frac{1}{2})| \leq Lc$, and equation 5 yields a contribution at most $4\beta L_{\max}\gamma\,L\,c$. On the far region, use (G1) and equation 5 to bound $|w\phi| \leq 4\beta L_{\max}\gamma\,W$, and Chebyshev to bound the measure by ${\rm PVar}_\theta[x]/c^2$. Summing gives $\|\nabla_\theta \mathcal{L}(x)\| \leq 4\beta L_{\max}\gamma\,(Lc + W\,{\rm PVar}_\theta[x]/c^2)$. Optimizing at $c = (W\,{\rm PVar}_\theta[x]/L)^{1/3}$ yields the stated $8\,\beta L_{\max}\gamma\,L^{2/3}W^{1/3}\,{\rm PVar}_\theta[x]^{1/3}$ bound.

**Proof sketch of Proposition E.6.** Decompose $\langle -\nabla\mathcal{L}, u_x \rangle$ into three parts: $\mathcal{I}_{\rm ring} + \mathcal{I}_{\rm near} + \mathcal{I}_{\rm end}$. On the ring $\mathcal{R}_{\varepsilon,\delta_{\rm ring}}$, by Assumption E.3, $-\bar{w}(p)\,{\rm sign}(p - \frac{1}{2}) \geq \tilde{\delta}_{\varepsilon,\delta_{\rm ring}}|p - \frac{1}{2}|$, and by Assumption E.4,

$$\mathbb{E}\big[\langle \phi, u_x \rangle\,{\rm sign}(p - \tfrac{1}{2})\,|p - \tfrac{1}{2}|\,\mathbf{1}_{\mathcal{R}_{\varepsilon,\delta_{\rm ring}}}\big] \ \geq \ \kappa_{\varepsilon,\delta_{\rm ring}}\,{\rm M2}_{\varepsilon,\delta_{\rm ring}}[x].$$

Thus $\mathcal{I}_{\rm ring} \geq \tilde{\delta}_{\varepsilon,\delta_{\rm ring}}\kappa_{\varepsilon,\delta_{\rm ring}}{\rm M2}_{\varepsilon,\delta_{\rm ring}}[x]$. On the near-center set, by (G2) and equation 5, the magnitude is at most $4\beta L_{\max}\gamma\,L\,\varepsilon$. On the endpoint set, use (G1)+equation 5 for the magnitude and

Chebyshev with threshold $1/2 - \delta_{\mathrm{ring}}$ to get the measure bound $\mathrm{PVar}_\theta[x]/(1/2 - \delta_{\mathrm{ring}})^2$, hence $4\beta L_{\max}\gamma\,W\,\mathrm{PVar}_\theta[x]/(1/2 - \delta_{\mathrm{ring}})^2$. Summing gives the directional bound.

For SNR, by the law of total variance,

$$\mathrm{Var}(\langle w\phi, u_x\rangle) = \mathbb{E}\big[\,\mathrm{Var}(\langle w\phi, u_x\rangle \mid p_\theta, \phi)\,\big] + \mathrm{Var}\big(\mathbb{E}[\langle w\phi, u_x\rangle \mid p_\theta, \phi]\big).$$

Given $(p_\theta, \phi)$, only $Z$ is random and $\mathrm{Var}(w \mid p_\theta) \leq \Delta_\star^2/4$ while $|\bar{w}(p_\theta)| \leq W$; multiplying by $\|\phi\|_2^2 \leq (4\beta L_{\max}\gamma)^2$ from equation 5 gives $\sqrt{\mathrm{Var}(\langle w\phi, u_x\rangle)} \leq 4\,\beta\,L_{\max}\,\gamma\,\sqrt{W^2 + \Delta_\star^2/4}$, which implies the stated SNR bounds when combined with the directional numerator bound.

# G  ALGORITHM-AGNOSTIC GENERALITY: INSTANTIATIONS FOR SIMPO, KTO, AND ORPO

We make our "PVar $\Rightarrow$ effective updates" theory concrete for three widely used preference-learning objectives: **SimPO**, **KTO**, and **ORPO**. Their pairwise gradients admit the generic form

$$\nabla_\theta \mathcal{L}(x) = \mathbb{E}\big[w(p_\theta, Z)\,\phi(x; y_i, y_j; \theta)\big], \qquad \phi = \beta\big(\nabla_\theta \log \pi_\theta(y_i|x) - \nabla_\theta \log \pi_\theta(y_j|x)\big), \quad (6)$$

with $p_\theta = \sigma(s_\theta)$ and $s_\theta = \beta\big(\log \pi_\theta(y_i|x) - \log \pi_\theta(y_j|x)\big)$. All expectations are over independently sampled responses $(y_i, y_j) \sim \pi_\theta(\cdot|x)^{\otimes 2}$ (and, when $Z$ is random, also over $Z$ conditional on $(x, y_i, y_j)$). We reuse the bound $\|\phi(x; y_i, y_j; \theta)\|_2 \leq 4\,\beta\,L_{\max}\,\gamma(x; \theta)$ from equation 5.

**Calibration and directional excitability assumptions (self-contained).**     We model label generation via a *calibration map* $\rho : [0, 1] \to [0, 1]$: $\mathbb{E}[Z \mid p_\theta] = \rho(p_\theta)$ with $\rho(1/2) = 1/2$ and $\rho$ differentiable. Define $\bar{w}(p) := \mathbb{E}[w(p, Z) \mid p]$. We adopt (G1)–(G2) from Assumption E.2 and the ring-slope gap Assumption E.3, together with the directional excitability Assumption E.4 (with $\delta_{\mathrm{ring}}$ optional when no endpoint trimming is used).

## G.1  A GENERIC LEMMA FOR LOGISTIC-TYPE PAIRWISE LOSSES

Consider any smooth per-pair loss $\ell(s_\theta, Z)$ with link $p_\theta = \sigma(s_\theta)$ and define $w(p_\theta, Z) := \partial\ell/\partial s$. Then $\nabla_\theta \ell = w(p_\theta, Z)\,\phi$, hence equation 6 holds after taking expectations. Assume for all $p \in [0, 1]$ and $Z \in \{0, 1\}$:

  (G1) (*Boundedness*) $|w(p, Z)| \leq W$ for some $W < \infty$.

  (G2) (*Lipschitz conditional mean*) $|\bar{w}(p) - \bar{w}(1/2)| \leq L\,|p - 1/2|$ near $1/2$.

  (G3) (*Monotone slope gap on $\mathcal{R}_\varepsilon$*) $\bar{w}'(p)$ has a fixed sign and $|\bar{w}'(p)| \geq \tilde{\delta}$ for all $|p - 1/2| \geq \varepsilon$.

For the logistic cross-entropy–type losses below we have $\bar{w}(p) = c \cdot \big(p - \rho(p)\big)$ for some $c > 0$, so that (G2) and (G3) follow from Assumption E.1 with $L = c \sup_p |1 - \rho'(p)|$ and $\tilde{\delta} = c\,\delta$, where $\delta$ is the slope-gap of $1 - \rho'$.[1]

**Proposition G.1** (Generic PVar upper bound for logistic-type losses). *Under (G1)–(G2) and the score bound equation 5, for any prompt $x$ we have*

$$\big\|\nabla_\theta \mathcal{L}(x)\big\| \leq 8\,\beta\,L_{\max}\,\gamma(x; \theta)\,\left(L^{2/3}W^{1/3}\right)\,\mathrm{PVar}_\theta[x]^{1/3}.$$

**Per-prompt SNR.**     For $g := w(p_\theta, Z)\,\phi$, define

$$\mathrm{SNR}(x) := \frac{\big|\langle\mathbb{E}[g],\,u_x\rangle\big|}{\sqrt{\mathrm{Var}(\langle g,\,u_x\rangle)}}.$$

**Proposition G.2** (Generic directional and SNR lower bounds — trimmed). *Let $\Delta_\star(p) := w(p, 1) - w(p, 0)$ and $\Delta_\star := \sup_p |\Delta_\star(p)|$. Assume (G3) and Assumption E.4 hold, and $\|\phi\| \leq 4\beta L_{\max}\gamma$.*

---

[1] When endpoint trimming is used, the conclusions remain valid with an additional endpoint slack term as in Proposition E.6.

*Then for any prompt $x$ and any inner trimming radius $\varepsilon \in (0, 1/2)$,*

$$\left\langle -\nabla_\theta \mathcal{L}(x),\, u_x \right\rangle \;\geq\; \tilde{\delta}\,\kappa_\varepsilon\,\mathrm{M2}_\varepsilon[x] \;-\; 4\,\beta\,L_{\max}\,\gamma(x;\theta)\,L\,\varepsilon,$$

$$\mathrm{SNR}(x) \;\geq\; \frac{\tilde{\delta}\,\kappa_\varepsilon\,\mathrm{M2}_\varepsilon[x] - 4\,\beta\,L_{\max}\,\gamma\,L\,\varepsilon}{4\,\beta\,L_{\max}\,\gamma(x;\theta)\,\sqrt{W^2 + \frac{1}{4}\Delta_\star^2}} \;\geq\; \frac{\tilde{\delta}\,\kappa_\varepsilon\,\mathrm{M2}_\varepsilon[x] - 4\,\beta\,L_{\max}\,\gamma\,L\,\varepsilon}{4\sqrt{2}\,W\,\beta\,L_{\max}\,\gamma(x;\theta)}.$$

*If an outer trimming radius $\delta_{\mathrm{ring}} \in (0, 1/2)$ is also used, add the endpoint slack $-4\,\beta\,L_{\max}\,\gamma\,W\,\mathrm{PVar}_\theta[x]/(1/2 - \delta_{\mathrm{ring}})^2$ to the numerators.*

## G.2 SIMPO

**Loss and gradient.** SimPO uses the pairwise logistic cross-entropy on $s_\theta = \beta(\log \pi_\theta(y_i|x) - \log \pi_\theta(y_j|x))$:

$$\ell_{\mathrm{SimPO}}(s_\theta, Z) \;=\; -\Big( Z \log p_\theta + (1-Z)\log(1-p_\theta) \Big), \qquad p_\theta = \sigma(s_\theta),$$

hence

$$\frac{\partial \ell_{\mathrm{SimPO}}}{\partial s}(s_\theta, Z) \;=\; p_\theta - Z, \qquad \nabla_\theta \ell_{\mathrm{SimPO}} \;=\; (p_\theta - Z)\,\phi.$$

Therefore $w_{\mathrm{SimPO}}(p, Z) = p - Z$.

**Verifying (G1)–(G3).** Boundedness: $|w| \leq 1$ so $W = 1$. Mean function: $\bar{w}(p) = \mathbb{E}[p - Z \mid p] = p - \rho(p)$, hence $L = \sup_p |1 - \rho'(p)|$ and $\bar{w}'(p) = 1 - \rho'(p)$; on a trimmed region, $|\bar{w}'(p)| \geq \delta$ with a fixed sign if $\rho'(1/2) > 1$ locally, so $\tilde{\delta} = \delta$.

**Consequences.** By Propositions G.1–G.2,

$$\|\nabla \mathcal{L}_{\mathrm{SimPO}}(x)\| \;\leq\; 8\,\beta\,L_{\max}\,\gamma(x;\theta)\left( \sup_p |1 - \rho'(p)| \right)^{2/3} \mathrm{PVar}_\theta[x]^{1/3},$$

and, for SNR (inner trimming only),

$$\mathrm{SNR}_{\mathrm{SimPO}}(x) \;\geq\; \frac{\delta\,\kappa_\varepsilon\,\mathrm{M2}_\varepsilon[x] - 4\,\beta\,L_{\max}\,\gamma\,L\,\varepsilon}{4\,\beta\,L_{\max}\,\gamma(x;\theta)\,\sqrt{1 + \frac{1}{4}}} \;\geq\; \frac{\delta\,\kappa_\varepsilon\,\mathrm{M2}_\varepsilon[x] - 4\,\beta\,L_{\max}\,\gamma\,L\,\varepsilon}{4\sqrt{2}\,\beta\,L_{\max}\,\gamma(x;\theta)}.$$

## G.3 KTO (SYMMETRIC CLASS-WEIGHTED LOGISTIC)

KTO is often implemented as a *class-weighted* logistic objective with a constant $\lambda > 0$:

$$\ell_{\mathrm{KTO}}(s_\theta, Z) \;=\; \lambda\Big( -Z \log p_\theta - (1-Z)\log(1-p_\theta) \Big).$$

Then

$$\frac{\partial \ell_{\mathrm{KTO}}}{\partial s}(s_\theta, Z) = \lambda\,(p_\theta - Z), \qquad \nabla_\theta \ell_{\mathrm{KTO}} = \lambda\,(p_\theta - Z)\,\phi,$$

so $w_{\mathrm{KTO}}(p, Z) = \lambda\,(p - Z)$.

**Verifying (G1)–(G3).** $W = \lambda$. Moreover $\bar{w}(p) = \lambda(p - \rho(p))$ implies $L = \lambda \sup_p |1 - \rho'(p)|$ and $\tilde{\delta} = \lambda\delta$.

**Consequences.**

$$\|\nabla \mathcal{L}_{\mathrm{KTO}}(x)\| \;\leq\; 8\,\beta\,L_{\max}\,\gamma(x;\theta)\,\lambda\left( \sup_p |1 - \rho'(p)| \right)^{2/3} \mathrm{PVar}_\theta[x]^{1/3}.$$

For SNR, the factor $\lambda$ cancels in numerator and denominator, yielding the same lower bound as SimPO:

$$\mathrm{SNR}_{\mathrm{KTO}}(x) \;\geq\; \frac{\delta\,\kappa_\varepsilon\,\mathrm{M2}_\varepsilon[x] - 4\,\beta\,L_{\max}\,\gamma\,L\,\varepsilon}{4\,\beta\,L_{\max}\,\gamma(x;\theta)\,\sqrt{1 + \frac{1}{4}}} \;\geq\; \frac{\delta\,\kappa_\varepsilon\,\mathrm{M2}_\varepsilon[x] - 4\,\beta\,L_{\max}\,\gamma\,L\,\varepsilon}{4\sqrt{2}\,\beta\,L_{\max}\,\gamma(x;\theta)}.$$

**Remark (from symmetric constants to general prospect weights).** If one uses $p$-dependent positive weights $c_+(p)$ and $c_-(p)$ in

$$\ell = -c_+(p)\, Z \log p \, - \, c_-(p)\,(1-Z) \log(1-p),$$

with $c_\pm$ bounded away from $0$ and $\infty$ and Lipschitz on $[0,1]$, then by the chain rule

$$\frac{\partial \ell}{\partial s} = -c_+(p)\, Z(1-p) + c_-(p)\,(1-Z)p + p(1-p)\Big(-c_+'(p)\, Z \log p - c_-'(p)\,(1-Z)\log(1-p)\Big).$$

Taking the conditional mean in $Z$ and subtracting the baseline $\bar{w}(1/2)$ gives $|\bar{w}(p) - \bar{w}(1/2)| \leq C\,|p - 1/2|$ on any band around $1/2$, where $C$ depends only on $\sup |c_\pm|$, $\inf c_\pm$, $\sup |c_\pm'|$, and $\sup |\rho'|$ (using that $p(1-p)$ is bounded and $\log p$ is Lipschitz on compact subintervals of $(0,1)$). Thus (G2) holds with $L = C$, while (G3) holds on $\mathcal{R}_\varepsilon$ with a reduced gap $|\bar{w}'(p)| \geq \underline{c}\,\delta - C'\varepsilon$ (for some $C' > 0$ depending on $\sup |c_\pm'|$); hence the same $1/3$-rate upper bound follows, with constants depending on $(c_\pm, \rho)$.

## G.4 ORPO

ORPO combines a *pairwise odds-ratio* term with an *SFT* term on the winner. At the pair level, the odds-ratio term is a logistic cross-entropy:

$$\ell_{\mathrm{OR}}(s_\theta, Z) = \lambda\Big(-Z \log p_\theta - (1-Z)\log(1-p_\theta)\Big), \qquad \lambda > 0,$$

so $w_{\mathrm{OR}}(p, Z) = \lambda\,(p - Z)$ and $\nabla_\theta \ell_{\mathrm{OR}} = \lambda\,(p_\theta - Z)\,\phi$. The full ORPO objective adds an SFT loss on the winner with weight $\mu \geq 0$:

$$\ell_{\mathrm{ORPO}} = \ell_{\mathrm{OR}} + \mu\, \ell_{\mathrm{SFT}}, \qquad \ell_{\mathrm{SFT}}(x; y_w) = -\log \pi_\theta(y_w | x).$$

Hence

$$\nabla_\theta \ell_{\mathrm{ORPO}}(x) = \underbrace{\lambda\, \mathbb{E}\big[(p_\theta - Z)\,\phi\big]}_{\text{pairwise odds-ratio term}} + \underbrace{\mu\, \mathbb{E}\big[-\nabla_\theta \log \pi_\theta(y_w|x)\big]}_{\text{SFT term}}.$$

**Pairwise term.** Exactly as in KTO (with $\lambda$), the pairwise component satisfies (G1)–(G3) with $W = \lambda$, $L = \lambda \sup_p |1 - \rho'(p)|$, and $\tilde{\delta} = \lambda\delta$. Therefore, by Propositions G.1–G.2,

$$\Big\|\lambda\, \mathbb{E}\big[(p_\theta - Z)\,\phi\big]\Big\| \leq 8\,\beta\, L_{\max}\, \gamma(x;\theta)\, \lambda\, \Big(\sup_p |1 - \rho'(p)|\Big)^{2/3} \mathrm{PVar}_\theta[x]^{1/3},$$

and the directional/SNR conclusions match those of KTO (for the pairwise component).

**SFT term.** By equation 5 with one response, $\|\nabla_\theta \log \pi_\theta(y_w|x)\|_2 \leq 2\, L_{\max}\, \gamma(x;\theta)$, hence

$$\Big\|\mu\, \mathbb{E}\big[-\nabla_\theta \log \pi_\theta(y_w|x)\big]\Big\|_2 \leq 2\,\mu\, L_{\max}\, \gamma(x;\theta).$$

This one-sided term does not depend on the loser $y_l$ and thus is *insensitive* to PVar.

## G.5 SUMMARY OF CONSTANTS AND TAKEAWAYS

All three methods have pairwise gradients of the form equation 6 with

$$w_{\mathrm{SimPO}}(p, Z) = p - Z, \qquad w_{\mathrm{KTO}}(p, Z) = \lambda\,(p - Z), \qquad w_{\mathrm{OR}}(p, Z) = \lambda\,(p - Z),$$

so that for the pairwise parts

$$(W, L, \tilde{\delta}) = \big(1,\, \sup_p |1 - \rho'(p)|,\, \delta\big) \text{ (SimPO)}, \qquad (W, L, \tilde{\delta}) = \big(\lambda,\, \lambda \sup_p |1 - \rho'(p)|,\, \lambda\delta\big) \text{ (KTO/OR)}.$$

Consequently, under Assumptions E.1, E.2, E.3, E.4,

**Upper bounds:** $\|\nabla \mathcal{L}_{\mathrm{SimPO}}(x)\| \leq 8\,\beta\, L_{\max}\, \gamma \Big(\sup_p |1 - \rho'(p)|\Big)^{2/3} \mathrm{PVar}_\theta[x]^{1/3},$

$$\|\nabla \mathcal{L}_{\mathrm{KTO}}(x)\| \leq 8\,\beta\, L_{\max}\, \gamma\, \lambda \Big(\sup_p |1 - \rho'(p)|\Big)^{2/3} \mathrm{PVar}_\theta[x]^{1/3},$$

$$\|\nabla \mathcal{L}_{\mathrm{ORPO}}(x)\| \leq 8\,\beta\, L_{\max}\, \gamma\, \lambda \Big(\sup_p |1 - \rho'(p)|\Big)^{2/3} \mathrm{PVar}_\theta[x]^{1/3} + 2\,\mu\, L_{\max}\, \gamma.$$

**Directional/SNR lower bounds (pairwise components, inner trimming):**

$$\langle -\nabla \mathcal{L}_{\text{SimPO}}, u_x \rangle \geq \delta \, \kappa_\varepsilon \, \text{M2}_\varepsilon[x] \; - \; 4 \, \beta \, L_{\max} \, \gamma \, L \, \varepsilon,$$

$$\langle -\nabla \mathcal{L}_{\text{KTO}}, u_x \rangle \geq \lambda \, \delta \, \kappa_\varepsilon \, \text{M2}_\varepsilon[x] \; - \; 4 \, \beta \, L_{\max} \, \gamma \, (\lambda \sup |1 - \rho'|) \, \varepsilon,$$

$$\text{SNR}_{\text{SimPO}}(x), \; \text{SNR}_{\text{KTO}}(x) \geq \frac{\delta \, \kappa_\varepsilon \, \text{M2}_\varepsilon[x] \; - \; 4 \, \beta \, L_{\max} \, \gamma \, L \, \varepsilon}{4 \, \beta \, L_{\max} \, \gamma \, \sqrt{1 + \frac{1}{4}}} \; \geq \; \frac{\delta \, \kappa_\varepsilon \, \text{M2}_\varepsilon[x] \; - \; 4 \, \beta \, L_{\max} \, \gamma \, L \, \varepsilon}{4\sqrt{2} \, \beta \, L_{\max} \, \gamma}.$$

**Key message.** For SimPO, KTO (symmetric weighting), and the pairwise component of ORPO, the *same* PVar-based theory applies *without change* up to a multiplicative constant that rescales the gradient. Thus, high-PVar prompts *necessarily* yield larger, directionally aligned, and higher-SNR updates across these algorithms, explaining the robust empirical gains observed when selecting data by PVar.

## H  ADDITIONAL EXPERIMENTAL RESULTS

### H.1  GENERALIZATION TO OTHER PREFERENCE OPTIMIZATION OBJECTIVES

Our theoretical framework in Section E posits that PVar controls the gradient norm and signal-to-noise ratio for a broad class of pairwise preference objectives, not just DPO. To validate this generalization, we extended our experimental analysis to include **SimPO** (Meng et al., 2024), **KTO** (Ethayarajh et al., 2024), and **ORPO** (Hong et al., 2024).

We trained Llama-3.1-8B-Instruct on the UltraFeedback dataset using the same PVar-based selection strategy (PVar Top 50%, Random 50%, PVar Bottom 50%). As shown in Table 5, the results consistently demonstrate that training on high-PVar prompts yields superior performance across all three algorithms.

| Algorithm | Selection Strategy | LC Win Rate (%) | Win Rate (%) |
|---|---|---|---|
| SimPO | PVar Top 50% | **39.1** | **44.0** |
| | Random 50% | 38.1 | 43.8 |
| | PVar Bottom 50% | 37.0 | 42.5 |
| KTO | PVar Top 50% | **40.0** | **45.6** |
| | Random 50% | 39.2 | 43.8 |
| | PVar Bottom 50% | 37.6 | 42.9 |
| ORPO | PVar Top 50% | **40.3** | **45.1** |
| | Random 50% | 38.8 | 44.3 |
| | PVar Bottom 50% | 38.2 | 43.8 |

Table 5: Generalization of PVar-based selection across different preference optimization algorithms (SimPO, KTO, ORPO).

### H.2  VALIDATION OF GRADIENT NORM DYNAMICS

A core theoretical contribution of this work (Theorem 4.1) is the upper bound on the gradient norm imposed by PVar. To empirically verify whether this bound translates to actual optimization dynamics, we tracked the gradient norms during the training of the Llama-3.1-8B-Instruct model on Chatbot Arena Conversations.

Figure 4 visualizes the gradient norm trajectories for models trained on Top, Random, and Bottom PVar subsets. The plot reveals two critical insights:

1. **Direct Verification of Signal Strength:** At the beginning of training, we observe a clear hierarchy: **Top-PVar (Green) > Random (Red) > Bottom-PVar (Blue)**. The Top-PVar model initiates with a significantly higher gradient norm compared to the Bottom-PVar model. This provides direct physical evidence that high-PVar prompts generate larger, more impactful gradient updates, validating Theorem 4.1 in practice.

2. **Evidence of Efficient Convergence:** The Top-PVar curve decays rapidly after the initial phase, indicating **fast convergence**. The strong initial signal allows the model to quickly satisfy preference constraints, driving the gradients toward zero. In stark contrast, the Bottom-PVar curve remains flat and high, signifying **optimization stagnation**. Due to the weak signal, the model struggles to reach the optimum, explaining the slower loss convergence observed in Figure 2.

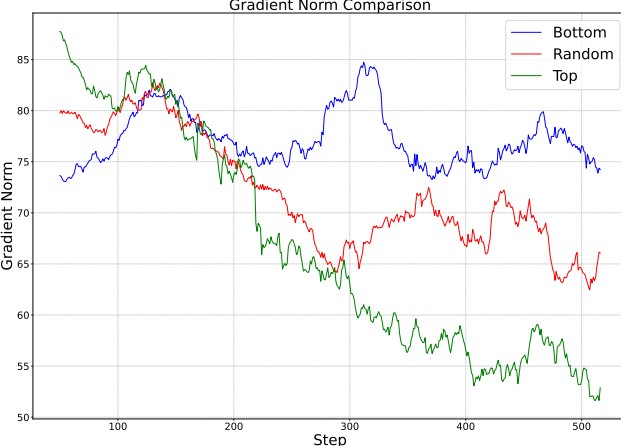

Figure 4: Gradient Norm Dynamics during DPO training. The **PVar Top** subset (green) starts with the highest gradient magnitude, indicating strong learning signals, and decays rapidly, reflecting efficient convergence. The **PVar Bottom** subset (blue) starts low and stagnates, indicating a struggle to optimize.

### H.3   PVAR VS. REWARD GAP: CORRELATION AND ROBUSTNESS

While PVar and the absolute Reward Gap (i.e., $|r(y_w) - r(y_l)|$) are related, they are fundamentally distinct metrics. PVar incorporates the non-linear transformation of the sigmoid function, prioritizing samples near the decision boundary where the model is "uncertain" (probability $\approx 0.5$), whereas Reward Gap scales linearly.

**Correlation Analysis.**   We analyzed the relationship between PVar and the mean absolute Reward Gap across the Chatbot Arena dataset. As shown in Figure 5, while there is a positive correlation, the relationship is non-linear.

- **Pearson correlation (linear):** $r = 0.63$
- **Spearman correlation (rank):** $\rho = 0.86$

The Spearman correlation is higher, indicating a monotonic relationship, but the lower Pearson correlation highlights that PVar captures distributional information that a simple linear gap misses. Specifically, PVar prevents the selection of "obvious" pairs (where the reward gap is massive, but gradient signal might be saturated) by naturally weighting variance.

### H.4   COMPARISON WITH OTHER DATA FILTERING BASELINES

To further demonstrate the effectiveness of PVar, we compared it against two filtering strategies:

**Perplexity (PPL).** Drawing inspiration from approaches that estimate sample difficulty (Kutalev & Markoff, 2024), we select prompts with the highest perplexity under the reference model. For a prompt $x$ of length $T$, we calculate:

$$\text{PPL}(x) = \exp\left(-\frac{1}{T}\sum_{t=1}^{T}\log P_{\text{ref}}(x_t \mid x_{<t})\right). \tag{7}$$

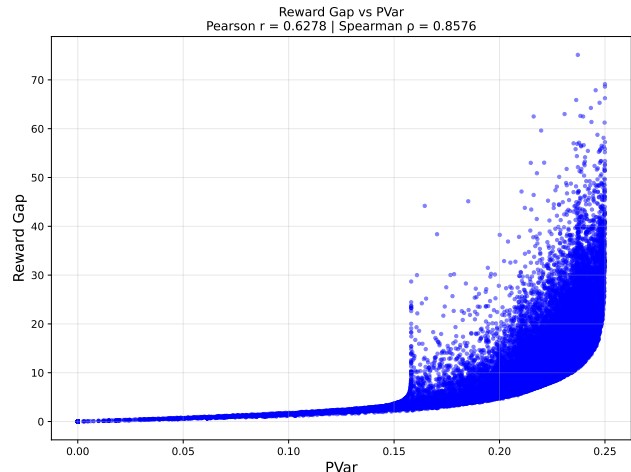

Figure 5: Scatter plot of Preference Variance (PVar) versus Absolute Reward Gap.

**Coreset Selection.** We employ a diversity-based strategy adapted from (Gupta et al.), where we embed all prompts into a feature space using the reference model. We then perform K-Means clustering to partition the dataset into $K$ clusters (where $K$ is the target budget). For each cluster $j$ with centroid $\mu_j$, we select the sample closest to the centroid:

$$x_j^* = \underset{x \in \mathcal{C}_j}{\operatorname{argmin}} \|\phi(x) - \mu_j\|_2, \tag{8}$$

where $\phi(x)$ is the embedding of prompt $x$. This ensures the selected subset maximizes semantic coverage.

We conducted experiments on the Chatbot Arena dataset using both Llama-3.1-8B-Instruct and Mistral-7B-Instruct-v0.2, selecting the top 50% of data for each method. As shown in Table 6, both baselines offer improvements over random selection but exhibit different strengths. However, **PVar Top 50%** consistently outperforms both, demonstrating that preference disagreement provides a more direct and unified signal for alignment than either difficulty or diversity alone.

| Base Model | Selection Strategy | AlpacaEval 2.0 | | Arena-Hard |
|---|---|---|---|---|
| | | **LC (%)** | **WR (%)** | **WR (%)** |
| Llama 3.1-8B-Instruct | Random 50% | 33.6 | 39.3 | 28.8 |
| | Coreset 50% | 34.2 | 39.5 | 29.2 |
| | Perplexity Top 50% | 35.0 | 39.4 | 29.6 |
| | **PVar Top 50%** | **36.2** | **39.6** | **30.0** |
| Mistral-7B-Instruct-v0.2 | Random 50% | 29.1 | 31.6 | 18.0 |
| | Coreset 50% | 30.8 | 33.0 | 18.6 |
| | Perplexity Top 50% | 31.2 | 32.7 | 19.1 |
| | **PVar Top 50%** | **32.5** | **34.5** | **20.4** |

Table 6: Comparison of PVar selection against Perplexity-based and Coreset (diversity-based) selection on Chatbot Arena dataset.

### H.5 SAMPLE EFFICIENCY ANALYSIS

To investigate the data efficiency of our method across different data budgets, we evaluated the performance of Llama-3.1-8B-Instruct trained on varying fractions of the Chatbot Arena dataset (10%, 30%, 50%). Table 7 compares the PVar Top selection against Random selection.

The results show that PVar Top selection consistently yields higher win rates than random selection at every data fraction. Notably, the model trained on just 30% of the PVar Top data achieves performance

comparable to the model trained on significantly larger randomly selected datasets, highlighting the sample efficiency gains enabled by our method.

| Data Fraction | Random Selection | | PVar Top Selection | |
|---|---|---|---|---|
| | LC (%) | WR (%) | LC (%) | WR (%) |
| 10% | 28.0 | 30.5 | **30.5** | **32.1** |
| 30% | 31.5 | 34.2 | **34.8** | **36.5** |
| 50% | 33.6 | 39.3 | **36.2** | **39.6** |

Table 7: Sample Efficiency Analysis. Llama-3.1-8B-Instruct trained on Chatbot Arena subsets of varying sizes.

### H.6 HYPERPARAMETER ROBUSTNESS

To ensure that the effectiveness of PVar is not dependent on specific hyperparameter configurations, we conducted ablation studies on the learning rate and the DPO $\beta$ parameter. Experiments were performed on the Chatbot Arena dataset using Llama-3.1-8B-Instruct, selecting the Top 50% of prompts. We compare PVar against Random selection as well as Perplexity and Coreset baselines to demonstrate its robustness.

**Learning Rate.** We tested three learning rates: $1 \times 10^{-7}$, $5 \times 10^{-7}$ (default), and $1 \times 10^{-6}$. As shown in Table 8, PVar Top selection consistently outperforms Random, Perplexity, and Coreset baselines across all learning rates.

**DPO Beta ($\beta$).** We also evaluated sensitivity to the KL-penalty coefficient $\beta$, testing values of 0.01, 0.05, and 0.1. The results confirm that while the absolute performance varies with $\beta$, the advantage of PVar remains robust, consistently achieving the highest LC Win Rates.

### H.7 ABLATION ON NUMBER OF SAMPLING RESPONSES ($N$)

Our main experiments estimate PVar using $N = 5$ generated responses per prompt. To assess the sensitivity of our method to this hyperparameter, we conducted an ablation study with $N = 3$, $N = 5$, and $N = 8$. We performed data selection on Chatbot Arena and trained Llama-3.1-8B-Instruct on the Top 50% subsets.

Table 9 demonstrates that the performance is robust to the choice of $N$. Even with $N = 3$, the selected subset significantly outperforms the Random baseline (33.6% LC), increasing $N$ to 8 provides marginal gains.

### H.8 GENERALIZATION TO CODE GENERATION

To further demonstrate the versatility of our approach beyond conversational tasks, we extended our evaluation to the domain of code generation. We fine-tuned the **Llama-3.1-8B-Instruct** model on the **MBPP** Austin et al. (2021) (Mostly Basic Python Problems) dataset and evaluated the performance on the **HumanEval** Chen et al. (2021) benchmark using the Pass@1 metric.

For this experiment, preference pairs were constructed based on test case execution results, and PVar was estimated accordingly. We compared our Top-PVar selection strategy against Random, Bottom-PVar, Perplexity, and Coreset baselines.

As detailed in Table 10, the model trained on the Top 50% PVar subset achieved the highest Pass@1 score of **68.3%**, outperforming the Random baseline (63.4%) and the Bottom subset (61.6%). Notably, PVar also surpassed both Coreset (64.2%) and Perplexity (65.5%) selection.

### H.9 CROSS-MODEL TRANSFERABILITY OF PVAR SELECTION

To investigate whether the effectiveness of PVar is model-specific or transferable, we conducted a cross-model selection experiment. Specifically, we used Llama-3.1-8B-Instruct to compute PVar and

| Hyperparameter | Value | Selection Strategy | LC Win Rate (%) |
|---|---|---|---|
| Learning Rate | $1 \times 10^{-7}$ | Random 50% | 32.4 |
| | | Perplexity Top 50% | 33.0 |
| | | Coreset 50% | 33.5 |
| | | **PVar Top 50%** | **34.8** |
| | $5 \times 10^{-7}$ | Random 50% | 33.6 |
| | | Perplexity Top 50% | 35.0 |
| | | Coreset 50% | 34.2 |
| | | **PVar Top 50%** | **36.2** |
| | $1 \times 10^{-6}$ | Random 50% | 32.9 |
| | | Perplexity Top 50% | 33.9 |
| | | Coreset 50% | 34.1 |
| | | **PVar Top 50%** | **35.4** |
| DPO Beta ($\beta$) | 0.01 | Random 50% | 34.0 |
| | | Coreset 50% | 35.1 |
| | | Perplexity Top 50% | 35.3 |
| | | **PVar Top 50%** | **36.8** |
| | 0.05 | Random 50% | 33.8 |
| | | Coreset 50% | 34.8 |
| | | Perplexity Top 50% | 35.1 |
| | | **PVar Top 50%** | **36.5** |
| | 0.1 | Random 50% | 33.6 |
| | | Perplexity Top 50% | 34.5 |
| | | Coreset 50% | 34.8 |
| | | **PVar Top 50%** | **36.2** |

Table 8: Hyperparameter robustness analysis on Learning Rate and DPO Beta ($\beta$).

| Number of Samples ($N$) | LC Win Rate (%) | Win Rate (%) |
|---|---|---|
| $N = 3$ | 35.8 | 39.2 |
| $N = 5$ | 36.2 | 39.6 |
| $N = 8$ | 36.3 | 39.8 |

Table 9: Ablation study on the number of sampled responses ($N$) used to estimate PVar. Performance is robust across different values of $N$. Training conducted on PVar Top 50% of Chatbot Arena with Llama-3.1-8B-Instruct. Metrics are on AlpacaEval 2.0.

select the Top 50% of prompts from the Chatbot Arena dataset. We then used this subset to train a different model, Mistral-7B-Instruct-v0.2.

As shown in Table 11, the results on AlpacaEval 2.0 reveal an interesting dual nature of Preference Variance. First, the **Cross-Model (Llama-selected)** subset outperforms the **Random** baseline (LC Win Rate 31.2% vs. 29.1%), suggesting that PVar captures some *intrinsic difficulty* or ambiguity in prompts that is universal across models. Prompts that confuse Llama are likely to be information-rich for Mistral as well. However, the **Self-Model (Mistral-selected)** subset achieves the highest performance (LC Win Rate 32.5%). This indicates that PVar also captures *model-specific epistemic uncertainty*—the specific knowledge boundaries and weaknesses of the model being trained.

| Selection Strategy | HumanEval Pass@1 (%) |
|---|---|
| PVar Bottom 50% | 61.6 |
| Random 50% | 63.4 |
| Coreset 50% | 64.2 |
| Perplexity Top 50% | 65.5 |
| **PVar Top 50%** | **68.3** |

Table 10: Performance on the HumanEval benchmark after DPO training on MBPP.

| Filtering Model | Training Set Description | LC Win Rate (%) | Win Rate (%) |
|---|---|---|---|
| N/A | Random 50% | 29.1 | 31.6 |
| Llama-3.1-8B | Cross-Model PVar Top 50% | 31.2 | 33.1 |
| Mistral-7B-v0.2 | **Self-Model PVar Top 50%** | **32.5** | **34.5** |

Table 11: Cross-model transferability analysis. Mistral-7B-Instruct-v0.2 was trained on data selected by different strategies. Metrics are on AlpacaEval 2.0.

