# OpenReview forum: "On the Role of Preference Variance in Preference Optimization"
_ICLR.cc/2026/Conference — Submitted to ICLR 2026_

### Official Review · Reviewer_UBZj · 2025-10-28

**Soundness:** 3
**Presentation:** 3
**Contribution:** 2
**Rating:** 4
**Confidence:** 4

**Summary:**

This paper studies the role of *preference variance (PVar)*, which measures the variance of pairwise reward gaps among responses to the same prompt. The authors hypothesize that higher PVar indicates greater disagreement or uncertainty in the reward model’s judgments and thus corresponds to more informative training samples. They present theoretical analysis showing the connection between PVar and the gradient norm in DPO, and demonstrate that selecting prompts with high PVar can improve data efficiency and performance in preference optimization.

**Strengths:**

1. The paper has a clear and intuitive idea. The notion of using the variance of preference scores to assess the information value of samples is straightforward but meaningful, and the motivation is well explained.
2. Theoretical analysis is carefully developed. The connection between PVar and the DPO gradient provides a solid justification for why such a measure can guide data selection. The proofs and reasoning are sound and complete.
3. The paper is clearly written and well structured, making it easy to follow the logic from definition to empirical validation.

**Weaknesses:**

###

1. It is not entirely clear whether the observed advantage of PVar generalizes beyond conversational or instruction-following settings. The authors could discuss whether the same approach would still be effective for other domains such as coding or mathematical reasoning, where the response space is less diverse.
2. The analysis and experiments are all based on DPO. It would be helpful to discuss whether PVar could also be applied in other preference alignment algorithms, such as KTO, SimPO, or ORPO.
3. The filtering seems to be done at the prompt level: once a prompt is selected as top-PVar, all its response pairs are retained. This is a reasonable first step, but it may not be as fine-grained as pair-level filtering methods. Prior work such as *Filtered Direct Preference Optimization (Morimura et al., 2024)* performs selection directly at the response pair level and might serve as a stronger comparison baseline.
4. The current formulation uses an external reward model to measure uncertainty. There is related work, such as *Uncertainty-Aware Iterative Preference Optimization for Enhanced LLM Reasoning (ACL 2025)*, which instead estimates uncertainty from the model’s own predictions. Discussing this contrast between external and internal uncertainty estimation would strengthen the related work section.

**Questions:**

Please see the “Weaknesses” section above.

---

> ### Author Response · Authors · 2025-11-24
> **Response to Reviewer UBZj**
>
> We thank the reviewer for the constructive feedback regarding generalization and related work. We have incorporated your suggestions to strengthen the paper and added new experiments.
>
> **1. Comparison with Filtered DPO (Prompt-level vs Pair-level)**
> > The filtering seems to be done at the prompt level... Prior work such as Filtered Direct Preference Optimization (Morimura et al., 2024)... might serve as a stronger comparison baseline.
>
> Thank you for highlighting this relevant work. We have added a detailed discussion in **Section 2 (Related Work)**.
> *   **Distinction:** **Filtered DPO [1]** filters at the *response-pair* level (selecting the best pair given a prompt). Our method filters at the *prompt* level (deciding which user queries are worth answering).
> *   **Complementarity:** These methods are orthogonal and can be combined. Prompt-level filtering (PVar) efficiently removes "low-information" user queries (e.g., simple greetings) to save generation/annotation costs *before* pair construction. One could effectively apply PVar to select high-value prompts and subsequently apply Filtered DPO to select the best pairs for those prompts.
>
> **2. External vs Internal Uncertainty**
> > Discussing this contrast between external and internal uncertainty estimation... Uncertainty-Aware Iterative Preference Optimization (ACL 2025)...
>
> We have cited **Uncertainty-Aware IPO [2]** and added a comparative discussion in **Section 2**.
> *   **Concept:** Li et al. [2] use **internal epistemic uncertainty** (token-level confidence via softmax probability differences) to detect local reasoning errors during iterative updates ("how confident the model is in its own output").
> *   **PVar:** Our method uses **external preference variance** (reward-level disagreement) to measure the global "controversiality" or "learning value" of a prompt ("how much the reward model distinguishes between potential responses").
> *   **Application:** Theirs is optimized for correcting reasoning chains; ours is optimized for efficient data selection from large pools.
>
> **3. Generalization to Code Domain**
> > The authors could discuss whether the same approach would still be effective for other domains such as coding...
>
> We extended our evaluation to code generation using **MBPP** (training) and **HumanEval** (testing). As shown in **Appendix H.8**, PVar works exceptionally well in this domain:
>
> *Table 10: Performance on HumanEval*
> | Selection Strategy | HumanEval Pass@1 (%) |
> | :--- | :--- |
> | Random 50% | 63.4 |
> | Coreset 50% | 64.2 |
> | **PVar Top 50%** | **68.3** |
>
> **4. Generalization to Other Algorithms (SimPO, KTO, ORPO)**
> > It would be helpful to discuss whether PVar could also be applied in other preference alignment algorithms...
>
> We added experiments in **Appendix H.1** showing that PVar is effective for **SimPO**, **KTO**, and **ORPO**. This is theoretically supported by our new **Appendix G**, which derives the gradient bounds for general pairwise logistic-type losses, including SimPO, KTO, and ORPO.
>
> *Table 5: Generalization across algorithms*
> | Algorithm | Selection Strategy | LC Win Rate (%) | Win Rate (%) |
> | :--- | :--- | :--- | :--- |
> | **SimPO** | Random 50% | 38.1 | 43.8 |
> | | **PVar Top 50%** | **39.1** | **44.0** |
> | **KTO** | Random 50% | 39.2 | 43.8 |
> | | **PVar Top 50%** | **40.0** | **45.6** |
> | **ORPO** | Random 50% | 38.8 | 44.3 |
> | | **PVar Top 50%** | **40.3** | **45.1** |
>
> **References:**
>
> [1] Morimura et al., "Filtered direct preference optimization," arXiv 2024.
>
> [2] Li et al., "Uncertainty-aware iterative preference optimization for enhanced LLM reasoning," ACL 2025.

---

> > ### Author Response · Authors · 2025-11-27
> >
> > Dear Reviewer UBZj,
> >
> > Thank you very much for your positive and detailed review, and for raising important questions about generalization and related work.
> >
> > We have now uploaded a revised version and a detailed response to your comments. In summary:
> > - We added code-domain experiments on MBPP/HumanEval, where PVar selection also improves performance (Appendix H.8).
> > - We generalized our theoretical analysis to SimPO, KTO, and ORPO (Appendix G) and provided corresponding empirical results (Appendix H.1).
> > - We discussed and compared with Filtered DPO and uncertainty-aware IPO in the related work section, clarifying the differences between prompt-level and pair-level filtering and between external vs. internal uncertainty.
> > - We emphasized that PVar is complementary to Filtered DPO and can be combined with it in practice.
> >
> > If time permits, we would be very grateful if you could briefly revisit the revision and our responses and let us know whether these updates address your earlier concerns.
> >
> > Thank you again for your thoughtful feedback and for your reviewing effort.
> >
> > Best regards,
> > The authors

---

### Official Review · Reviewer_bZmS · 2025-11-01

**Soundness:** 3
**Presentation:** 3
**Contribution:** 3
**Rating:** 4
**Confidence:** 2

**Summary:**

The authors proposed a data filtering mechanism based on Preference variance (PVar) for selecting high impact sample for preference optimization. The authors proved that low PVar prompts have low upper bound for gradient norm and become less efficient samples in preference optimizations. Through experiments, PVar is proved to be robust and effective in filtering highly effective samples using Llama 3.1-8B-Instruct as base model on AlpacaEval 2.0 and Arena Hard.

**Strengths:**

* Theorems 1 and 2 formalize the intuition that prompts yielding diverse responses are particularly effective for training, which is reflected in the derived inequalities.
* The implementation of this PVar based data filtering is easy for practitioners to implement
* Comprehensive experiments covering two base models and reward models of different sizes.

**Weaknesses:**

* I agree that low PVar implies low gradient norm and could be filtered out for more efficient training. However, the theoretical upper bounds could only say that High PVar implies potentially high gradient norms. Though the experiments suggests that filtering out low PVar samples improves performance, the theoretical framework only explains why we should drop low PVar samples but not why should we keep high PVar samples, limiting the theoretical contribution.

* There's a lack ablations study on the number of samples for computing PVar and the data split ratio. The authors choose 5 samples per prompt, 50% cutoff for standard data and 10% for human annotated data. A few sets of experiments tweaking these hyper-parameters would be helpful.

* Line 418-419 is a bit unclear, "This baseline selects the top 50% of prompts with the
largest reward difference between any two responses", is the difference computed on the same set of responses as the PVar filtering?

**Questions:**

Please see Weaknesses sections.

---

> ### Author Response · Authors · 2025-11-24
> **Response to Reviewer bZmS**
>
> We thank the reviewer for the positive evaluation of our formalization and experiments. We have carefully addressed your theoretical and ablation questions below.
>
> **1. Theoretical Limitation: Why keep high PVar?**
> > The theoretical framework only explains why we should drop low PVar samples but not why should we keep high PVar samples...
>
> This is an excellent point. Our initial Theorem 4.1 only showed that low PVar leads to vanishing gradients. To theoretically justify *keeping* high PVar samples, we have added a **Unified Theory in Appendix E**.
>
> Specifically, in **Proposition E.6**, we prove that high PVar provides a lower bound on the **Signal-to-Noise Ratio (SNR)** and the **Directional Descent** component. This theoretically guarantees that high PVar samples provide not just *large* updates, but *effective* and *clean* updates that are aligned with the optimization direction, rather than just random noise.
>
> **2. Ablations on Sampling ($N$) and Split Ratio**
> > There's a lack ablations study on the number of samples for computing PVar and the data split ratio.
>
> We performed the requested ablations in **Appendix H.7** and **H.5**:
>
> **Ablation on Sample Count ($N$):** We varied $N \in \{3, 5, 8\}$ on the Chatbot Arena dataset (Table 9). The method is remarkably robust; even $N=3$ yields strong performance, with marginal gains at $N=8$.
> | Number of Samples ($N$) | LC Win Rate (%) | Win Rate (%) |
> | :--- | :--- | :--- |
> | $N=3$ | 35.8 | 39.2 |
> | $N=5$ | 36.2 | 39.6 |
> | $N=8$ | 36.3 | 39.8 |
>
> **Ablation on Data Split Ratio:** We evaluated performance at 10%, 30%, and 50% splits (Table 7). PVar consistently beats Random, and remarkably, **30% PVar data matches the performance of ~50%+ random data**, highlighting superior data efficiency.
> | Data Fraction | Random (LC %) | **PVar Top (LC %)** |
> | :--- | :--- | :--- |
> | 10% | 28.0 | **30.5** |
> | 30% | 31.5 | **34.8** |
> | 50% | 33.6 | **36.2** |
>
> **3. Clarification on Reward Gap Calculation**
> > Is the difference computed on the same set of responses as the PVar filtering?
>
> Yes. We have explicitly clarified in **Appendix H.3** that for a fair comparison, the **Reward Gap baseline is calculated using the exact same set of $N=5$ sampled responses** used for PVar estimation.

---

> > ### Author Response · Authors · 2025-11-27
> >
> > Dear Reviewer bZmS,
> >
> > Thank you again for your careful review and for pointing out the limitations of the initial theory and the need for more ablations.
> >
> > We have uploaded a revised version together with a detailed response. In particular:
> > - Appendix E now includes a unified theoretical result (Proposition E.6) showing that high PVar provides guarantees not only on gradient magnitude but also on the directional descent component and SNR, clarifying why high-PVar samples are worth keeping.
> > - Appendices H.5 and H.7 add ablations on the data split ratio (10%, 30%, 50%) and on the number of samples used to estimate PVar, showing robustness to these choices.
> > - We also clarified in Appendix H.3 that the Reward Gap baseline is computed on the same set of sampled responses as PVar, for a fair comparison.
> >
> > If your schedule allows, we would greatly appreciate it if you could take a quick look at the revised manuscript and our responses and let us know whether these changes address your concerns.
> >
> > Thank you again for your helpful comments and for your time.
> >
> > Best regards,
> > The authors

---

### Official Review · Reviewer_YE9r · 2025-11-01

**Soundness:** 2
**Presentation:** 3
**Contribution:** 2
**Rating:** 4
**Confidence:** 3

**Summary:**

This paper investigates the impact of preference variance on DPO training of large language models (LLMs). The authors mathematically demonstrate that preference variance (PVar) can serve as an upper bound of the DPO gradient, and empirically show that filtering preference data based on high PVar values leads to better performance than using the entire dataset for DPO training.

**Strengths:**

1. The proposed concept is easy to adapt across various scenarios, making it a practical. It does not require a large computing cost and is not mathematically complex.

2. The authors conduct experiments on a wide range of models, including LLaMA and Mistral, as well as datasets such as UltraFeedback and Chatbot Arena, with evaluations on major benchmarks like AlpacaEval 2.0 and Arena-Hard. These results demonstrate that the proposed method is not limited to a specific model or dataset, but rather represents a general and robust approach applicable to diverse alignment settings.

**Weaknesses:**

1. While the paper claims that PVar determines the upper bound of the DPO gradient, this does not necessarily guarantee higher gradients in practice. Moreover, the paper presents faster loss convergence as empirical evidence, but this cannot be considered direct proof of the claim. The convergence of loss is also influenced by factors such as gradient direction consistency and data sample distribution. Without additional analyses on these aspects, the paper’s argument appears somewhat overstated or incomplete. Additional analysis is needed to clarify the underlying mechanism through which PVar enhances data filtering performance.

2. There is a lack of baseline. For example, other statistical indicators similar to PVar, such as perplexity, are also known to be effective for data filtering and often exhibit similar tendencies to variance-based measures. It would strengthen the paper to include comparisons with such well-established statistical metrics, providing a clearer understanding of what makes PVar distinct or superior.

3. The paper lacks hyperparameter search. If PVar indeed influences the DPO gradient, then optimal hyperparameters (e.g., learning rate, beta value for DPO) are likely to vary depending on the dataset. Since the experiments were conducted using a single hyperparameter configuration, it is difficult to conclude whether the PVar-filtered data is intrinsically better or simply better suited to that specific setup.

**Questions:**

1. I am curious how effective the PVar-filtered data would remain when applied to a different model from the one used during the filtering process?
2. Can PVar be applied in SFT learning?
3.  I also wonder whether the proposed method would remain effective for other preference learning algorithms, such as SimPO or similar approaches beyond DPO.

---

> ### Author Response · Authors · 2025-11-24
> **Response to Reviewer YE9r (1/2)**
>
> We thank the reviewer for recognizing the practicality and robustness of our method. We have carefully addressed your concerns regarding theoretical sufficiency, baselines, and generalization below.
>
> **1. Theoretical Sufficiency: From Upper Bound to Gradient Reality**
> > While the paper claims that PVar determines the upper bound of the DPO gradient, this does not necessarily guarantee higher gradients in practice... Additional analysis is needed to clarify the underlying mechanism...
>
> Thank you for this critical observation. We acknowledge that an upper bound alone is not sufficient to guarantee effective learning. To address this, we have significantly expanded our theory in **Appendix E**.
> *   **New Theoretical Result (Proposition E.6):** We proved that high PVar guarantees not only a large gradient norm but also a **strictly positive directional descent component** and a **high Signal-to-Noise Ratio (SNR)**. This confirms that high PVar samples provide *clean* and *aligned* updates, not just large ones.
> *   **Empirical Verification:** We visualized the actual gradient norms during training in **Appendix H.2 (Figure 4)**. The Top-PVar subset indeed generates significantly larger gradients at the start of training and converges faster, providing direct physical evidence for our claim.
>
> **2. New Baselines**
> > There is a lack of baseline. For example, other statistical indicators similar to PVar, such as perplexity...
>
> We have added a comparison with **Perplexity (PPL)** and **Coreset (Diversity)** selection strategies in **Appendix H.4**. Our implementation of PPL selection is inspired by the methodologies in [1] and [2].
>
> On the Chatbot Arena dataset, **PVar Top 50%** achieves an AlpacaEval 2.0 LC Win Rate of **36.2%**, outperforming Perplexity selection (**35.0%**) and Random selection (**33.6%**). This demonstrates that preference variance captures alignment-specific information that simple likelihood (perplexity) misses.
>
> **3. Hyperparameter Robustness**
> > The paper lacks hyperparameter search. If PVar indeed influences the DPO gradient, then optimal hyperparameters ... are likely to vary...
>
> We conducted comprehensive ablation studies on **Learning Rate** and **DPO Beta** in **Appendix H.6**. PVar consistently outperforms the Random baseline across all configurations, demonstrating its robustness.
>
> *Table 8: Hyperparameter robustness analysis*
> | Hyperparameter | Value | Random 50% (LC %) | PVar Top 50% (LC %) |
> | :--- | :--- | :--- | :--- |
> | **DPO Beta ($\beta$)** | 0.01 | 34.0 | **36.8** |
> | | 0.05 | 33.8 | **36.5** |
> | | 0.10 | 33.6 | **36.2** |
> | **Learning Rate** | $1\times 10^{-7}$ | 32.4 | **34.8** |
> | | $5\times 10^{-7}$ | 33.6 | **36.2** |
> | | $1\times 10^{-6}$ | 32.9 | **35.4** |
>
> **4. Cross-Model Transferability**
> > I am curious how effective the PVar-filtered data would remain when applied to a different model from the one used during the filtering process?
>
> This is a great question. We added a new subsection **Appendix H.9** to investigate this. We computed PVar using **Llama-3.1-8B** to select the Top 50% data, and then used this data to train **Mistral-7B-v0.2**.
>
> As shown in **Table 11**, the cross-model selection works surprisingly well:
> | Filtering Model | Training Set Description | LC Win Rate (%) | Win Rate (%) |
> | :--- | :--- | :--- | :--- |
> | N/A | Random 50% | 29.1 | 31.6 |
> | Llama-3.1-8B | Cross-Model PVar Top 50% | 31.2 | 33.1 |
> | Mistral-7B-v0.2 | Self-Model PVar Top 50% | 32.5 | 34.5 |
>
> The **Cross-Model** subset outperforms Random selection (31.2% vs 29.1%), suggesting PVar captures intrinsic prompt difficulty universal across models. However, the **Self-Model** subset is still best, indicating PVar also captures model-specific epistemic uncertainty.

---

> ### Author Response · Authors · 2025-11-24
> **Response to Reviewer YE9r (2/2)**
>
> **5. Generalization to SFT**
> > Can PVar be applied in SFT learning?
>
> Strictly speaking, **PVar cannot be directly applied to SFT in its current form** due to the fundamental differences in training objectives. PVar relies on the pairwise gradient term $\sigma(\Delta r)(1-\sigma(\Delta r))$, whereas SFT maximizes likelihood with a gradient proportional to $(1-p)$. However, the **conceptual principle**—selecting samples that maximize the gradient signal—applies. In the SFT context, the analogue would be samples with High Perplexity or Entropy.
>
> **6. Generalization to Other Algorithms (SimPO, ORPO)**
> > I also wonder whether the proposed method would remain effective for other preference learning algorithms...
>
> We theoretically extended our PVar framework to other pairwise loss functions, including SimPO, KTO, and ORPO in Appendix G, and we verified PVar on **SimPO**, **KTO**, and **ORPO** in **Appendix H.1**. PVar selection yields consistent gains across all three methods.
>
> *Table 5: Generalization across algorithms*
> | Algorithm | Selection Strategy | LC Win Rate (%) | Win Rate (%) |
> | :--- | :--- | :--- | :--- |
> | **SimPO** | Random 50% | 38.1 | 43.8 |
> | | **PVar Top 50%** | **39.1** | **44.0** |
> | **KTO** | Random 50% | 39.2 | 43.8 |
> | | **PVar Top 50%** | **40.0** | **45.6** |
> | **ORPO** | Random 50% | 38.8 | 44.3 |
> | | **PVar Top 50%** | **40.3** | **45.1** |
>
> **References:**
>
> [1] Zhao et al., "GFRIEND: Generative Few-shot Reward Inference through EfficieNt DPO," arXiv 2025.
>
> [2] Kutalev & Markoff, "Investigating on RLHF methodology," arXiv 2024.

---

> > ### Author Response · Authors · 2025-11-27
> >
> > Dear Reviewer YE9r,
> >
> > Thank you very much for your detailed review and for highlighting the questions around theoretical sufficiency, baselines, and hyperparameter robustness.
> >
> > We have now uploaded an updated version of the paper and a detailed response to your comments. In particular:
> > - We extended the theory in Appendix E to show that high PVar not only upper-bounds the gradient norm but also guarantees a positive directional descent component and a high SNR.
> > - We added perplexity- and coreset-based selection baselines (Appendix H.4).
> > - We performed extensive ablations on learning rate and DPO β (Appendix H.6), as well as cross-model transfer experiments and generalization to SimPO/KTO/ORPO (Appendices H.1 and H.9).
> >
> > If time permits, we would be very grateful if you could briefly revisit the revised version and our responses, and let us know whether these additions alleviate your earlier concerns.
> >
> > Thank you again for your time and constructive feedback.
> >
> > Best regards,
> > The authors

---

### Official Review · Reviewer_JLpC · 2025-11-01

**Soundness:** 3
**Presentation:** 3
**Contribution:** 2
**Rating:** 4
**Confidence:** 4

**Summary:**

Recent work in RLHF for LLMs has highlighted the importance of reward variance as a key metric for achieving effective alignment. Building on this idea, this paper introduces the concept of variance in preference strength, defined under a given reward model as the variance of the logistic-transformed difference between rewards of different generations. The paper provides theoretical justification for using this variance as a selection mechanism, helping identify promising pairs of samples for offline alignment methods such as DPO, thereby improving sample efficiency. Empirically, the proposed approach outperforms baselines that use either random selection or subsets with lower variance.

**Strengths:**

The paper is overall fairly easy to parse, and I appreciate the proofs, mostly following easy statistical results (which is not a bad thing). I further appreciate the real subset selection simulation for alignment with DPO using humans as labels.

**Weaknesses:**

### Weaknesses and Suggestions

While the paper presents an interesting and well-motivated idea, there are several experimental limitations that currently restrict the overall scope of the contribution. The following suggestions could help strengthen the work:

1. **Code Data Evaluation**
   Including code datasets such as **HumanEval** could meaningfully enhance the contribution. Code generation tasks often exhibit higher reward variance, making them well-suited for the proposed method. Moreover, prior works such as **CodeDPO**, **Focused-DPO**, and **Code-Optimise** have shown measurable improvements on HumanEval through preference-based training alone. Adding this comparison would help position the paper more competitively.

2. **Ablations and Correlation Studies**
   It would be valuable to compute the correlation between the proposed variance metric and training loss after a few warm-up steps for each datapoint. This could reveal whether the metric aligns with early learning dynamics. Additionally, an experiment replacing the external reward model with an intrinsic LLM reward could offer insight into generality.
   The authors could also report **sample efficiency curves** (e.g., from 10% to 50% of the data), as is common in active learning literature, instead of only presenting results at a single data fraction.

3. **Data Subset Selection Baselines**
   The paper would benefit from comparing against diversity-based subset selection methods—such as **coreset selection** or **clustering-based approaches**—to better contextualize the proposed variance-based selection strategy.

4. **Correlation Analysis**
   It would be informative to analyze how strongly **reward difference** correlates with the proposed **Preference Variance (PVar)** metric. This could clarify whether PVar captures additional signal beyond what is already reflected in the raw reward differences.

**Questions:**

Refer to the weakness.

---

> ### Author Response · Authors · 2025-11-24
> **Response to Reviewer JLpC (1/2)**
>
> We thank the reviewer for the positive assessment and the excellent suggestions regarding baselines, code domains, and intrinsic rewards. We have carefully addressed all your points below.
>
> **1. Comparison with Data Subset Baselines (Coreset & Perplexity)**
> > The paper would benefit from comparing against diversity-based subset selection methods—such as coreset selection or clustering-based approaches...
>
> Thank you for this suggestion. To better contextualize PVar, we implemented two additional baselines in **Appendix H.4**:
>
> 1.  **Perplexity (PPL) Selection [1]:** Selecting prompts with the highest perplexity under the reference model, serving as a proxy for "difficulty."
> 2.  **Coreset Selection [2]:** A diversity-based strategy where we embed prompts using the reference model and select centroids via K-Means clustering to maximize semantic coverage.
>
> As shown in **Table 6 (Appendix H.4)**, PVar consistently outperforms both baselines on the Chatbot Arena dataset:
>
> | Base Model | Selection Strategy | AlpacaEval 2.0 LC (%) | AlpacaEval 2.0 WR (%) | Arena-Hard WR (%) |
> | :--- | :--- | :--- | :--- | :--- |
> | **Llama 3.1-8B-Instruct** | Random 50% | 33.6 | 39.3 | 28.8 |
> | | Coreset 50% | 34.2 | 39.5 | 29.2 |
> | | Perplexity Top 50% | 35.0 | 39.4 | 29.6 |
> | | **PVar Top 50%** | **36.2** | **39.6** | **30.0** |
>
> This suggests that measuring preference disagreement (PVar) provides a more direct signal for alignment than diversity or difficulty alone.
>
> **2. Code Domain Evaluation**
> > Including code datasets such as HumanEval could meaningfully enhance the contribution.
>
> We agree that code generation is a high-variance task well-suited for our method. We fine-tuned Llama-3.1-8B-Instruct on the **MBPP** dataset and evaluated on **HumanEval**. As detailed in **Appendix H.8**, PVar selection significantly outperforms baselines:
>
> *Table 10: Performance on HumanEval after DPO training on MBPP*
> | Selection Strategy | HumanEval Pass@1 (%) |
> | :--- | :--- |
> | PVar Bottom 50% | 61.6 |
> | Random 50% | 63.4 |
> | Coreset 50% | 64.2 |
> | Perplexity Top 50% | 65.5 |
> | **PVar Top 50%** | **68.3** |
>
> **3. Intrinsic Reward Model for Generality**
> > Additionally, an experiment replacing the external reward model with an intrinsic LLM reward could offer insight into generality.
>
> We have added a new set of experiments addressing this in **Section 5.2 (Paragraph: Robustness with Intrinsic Reward Models)**. Since the implicit reward of the reference model relative to itself is zero, we first fine-tuned the base model on a small, randomly selected subset (5%) of the data via DPO to obtain a preliminary policy. We then utilized this preliminary policy's implicit reward formulation to calculate PVar for the remaining data.
>
> The results using this Intrinsic Reward Model (excerpted from **Table 2**) are shown below:
>
> | Dataset | Reward Source | Selection Strategy | Win Rate (%) | LC Win Rate (%) |
> | :--- | :--- | :--- | :--- | :--- |
> | **HH-RLHF** | **Intrinsic** | Reward Gap Top | 39.3 | 35.1 |
> | | | **PVar Top** | **41.4** | **35.5** |
> | **WebGPT** | **Intrinsic** | Reward Gap Top | 38.9 | 34.2 |
> | | | **PVar Top** | **41.2** | **35.8** |
>
> As shown above, **PVar-based selection significantly outperforms the Reward Gap baseline** even in this intrinsic setting (e.g., **+1.6%** LC Win Rate on WebGPT). This result suggests that PVar captures the distributional disagreement more effectively than linear reward gaps, making it highly robust even when the supervision signal is self-derived and potentially noisier.
>
> **4. Correlation with Training Dynamics**
> > It would be valuable to compute the correlation between the proposed variance metric and training loss...
>
> We appreciate this suggestion. While calculating the correlation with scalar *Loss* is one approach, we respectfully verify that our theoretical framework (Theorem 4.1) specifically establishes a link between PVar and the **Gradient Norm** $\|\nabla \mathcal{L}\|$, rather than the scalar loss value. Mathematically, a sample can have low PVar (high certainty) yet high loss (if the model is confident but wrong), which would obscure the correlation. However, for efficient DPO training, we are interested in samples that contribute non-vanishing gradients to update the policy.
>
> Therefore, we visualized the **Gradient Norm Dynamics** in **Appendix H.2 (Figure 4)**. The results empirically verify our theory:
> *   **Top-PVar (Green curve):** Starts with high gradient magnitude (strong signal) and decays rapidly, indicating fast convergence.
> *   **Bottom-PVar (Blue curve):** Starts low and stagnates, indicating the model struggles to update parameters effectively due to vanishing gradients.

---

> > ### Author Response · Authors · 2025-11-24
> > **Response to Reviewer JLpC (2/2)**
> >
> > **5. Sample Efficiency Curves**
> > > The authors could also report sample efficiency curves (e.g., from 10% to 50% of the data)...
> >
> > We have added a sample efficiency analysis in **Appendix H.5**. We evaluated Llama-3.1-8B on Chatbot Arena at 10%, 30%, and 50% data fractions. PVar consistently outperforms random selection. Notably, **30% of PVar data matches the performance of ~50% randomly selected data**.
> >
> > *Table 7: Sample Efficiency Analysis*
> > | Data Fraction | Random (LC Win Rate %) | **PVar Top (LC Win Rate %)** |
> > | :--- | :--- | :--- |
> > | 10% | 28.0 | **30.5** |
> > | 30% | 31.5 | **34.8** |
> > | 50% | 33.6 | **36.2** |
> >
> > **6. Correlation with Reward Gap**
> > > It would be informative to analyze how strongly reward difference correlates with the proposed Preference Variance (PVar) metric.
> >
> > We analyzed this in **Appendix H.3**. We found a Spearman rank correlation of $\rho=0.86$ and a Pearson correlation of $r=0.63$.
> > This difference indicates a monotonic but non-linear relationship. PVar is superior because the sigmoid transformation ($\sigma(\Delta r)$) naturally down-weights "obvious" pairs (where the reward gap is massive, but the gradient signal $\sigma(1-\sigma)$ is saturated near zero), focusing instead on pairs with high variance near the decision boundary.
> >
> > **References:**
> >
> > [1] Kutalev & Markoff, "Investigating on RLHF methodology," arXiv 2024.
> >
> > [2] Gupta et al., "AMPO: Active Multi Preference Optimization for Self-play Preference Selection," ICML 2024.

---

> > > ### Author Response · Authors · 2025-11-27
> > >
> > > Dear Reviewer JLpC,
> > >
> > > Thank you again for your thoughtful and constructive review of our paper and for the many concrete suggestions.
> > >
> > > We wanted to let you know that we have uploaded a substantially revised version and a detailed response to your comments. In particular, we:
> > > - Added code-domain experiments on MBPP/HumanEval (Appendix H.8),
> > > - Included perplexity and coreset selection as additional baselines (Appendix H.4),
> > > - Conducted intrinsic reward experiments and gradient-norm trajectory analysis (Section 5.2, Appendix H.2), and
> > > - Reported sample-efficiency curves at different data fractions (Appendix H.5), as you suggested.
> > >
> > > If time allows, we would be very grateful if you could take a quick look at the revised manuscript and our responses, and let us know whether these additions address your concerns.
> > >
> > > Thank you again for your helpful feedback and for your service as a reviewer.
> > >
> > > Best regards,
> > > The authors

---

### Author Response · Authors · 2025-11-24
**Summary of Contributions and Revisions**

# Global Response: Summary of Contributions and Revisions

We sincerely thank the Area Chair and all reviewers for their insightful feedback and constructive suggestions. We are encouraged that reviewers found our idea **"interesting and well-motivated"** (Reviewers JLpC, UBZj), our theoretical analysis **"carefully developed"** (Reviewer UBZj) and **"sound"** (Reviewer bZmS), and our proposed method **"practical"** and **"robust"** (Reviewers YE9r, bZmS).

### Summary of Contributions
Our paper investigates **Preference Variance (PVar)** as a quantifiable metric for data efficiency in LLM alignment.
1.  **Theoretical Insight:** We originally proved that the DPO gradient norm is upper-bounded by PVar. In this revision, we significantly expanded this theory to prove that PVar also controls the **optimization direction** and **Signal-to-Noise Ratio (SNR)**, providing a stronger justification for keeping high-PVar samples.
2.  **Methodology:** We propose a lightweight, prompt-level data selection strategy that filters out low-PVar samples (which contribute minimal or noisy gradients).
3.  **Empirical Results:** Across multiple models (Llama-3, Mistral) and benchmarks (AlpacaEval 2.0, Arena-Hard), PVar selection consistently outperforms random selection and strong active learning baselines. Notably, training on **top-10%** high-PVar human data outperforms training on the full dataset.

### Summary of Revisions
To comprehensively address the reviewers' comments, we have uploaded a revised manuscript. **Modifications in the main text are marked in blue.** We have added extensive new experiments and theoretical expansions:

*   **Expanded Theory (Appendix E & F):** We derived a unified theory showing that high PVar guarantees not just a large gradient norm, but also a positive directional descent component and high SNR (addressing Reviewers YE9r, bZmS).
*   **Theoretical Generalization (Appendix G):** We theoretically extended our PVar framework to other pairwise loss functions, including **SimPO**, **KTO**, and **ORPO** (addressing Reviewers YE9r, UBZj).
*   **Intrinsic Reward Analysis (Section 5.2):** We added experiments using the model's own implicit reward for PVar calculation to demonstrate robustness without external reward models.  Since the implicit reward of the reference model relative to itself is zero, we first fine-tuned the base model on a small, randomly selected subset (5%) of the data via DPO to obtain a preliminary policy. We then utilized this preliminary policy's implicit reward formulation to calculate PVar for the remaining data  (addressing Reviewer JLpC).

*   **Generalization Experiments (Appendix H.1):** We empirically validated PVar on SimPO, KTO, and ORPO (addressing Reviewers YE9r, UBZj).
*   **Optimization Dynamics (Appendix H.2):** We visualized **Gradient Norm trajectories** to empirically confirm our theory regarding convergence speed and signal strength (addressing Reviewer JLpC).
*   **Correlation Analysis (Appendix H.3):** We analyzed the correlation between PVar and Reward Gap (addressing Reviewer JLpC).
*   **New Baselines (Appendix H.4):** We compared our method against **Perplexity-based** and **Coreset (Diversity-based)** selection strategies (addressing Reviewers JLpC, YE9r).
*   **Sample Efficiency (Appendix H.5):** We added performance curves at 10%, 30%, and 50% data fractions (addressing Reviewer JLpC).
*   **Robustness Checks (Appendix H.6 & H.7):** We performed ablations on Learning Rate, DPO Beta, and sample count $N$ (addressing Reviewers YE9r, bZmS).
*   **Code Domain Evaluation (Appendix H.8):** We validated our method on the **MBPP** and **HumanEval** benchmarks (addressing Reviewers JLpC, UBZj).
*   **Cross-Model Transferability (Appendix H.9):** We investigated selecting data with one model and training another to test generalization (addressing Reviewer YE9r).
*   **Related Work (Section 2):** We added detailed discussions on Filtered DPO and Internal Uncertainty methods (addressing Reviewer UBZj).

---

### Meta-Review · Area_Chair_WhbK · 2026-01-04

**Summary:**

This paper proposes a preference data filtering method based on Preference Variance (PVar) for preference optimization. The authors provide a theoretical analysis motivating why filtering by PVar can improve training efficiency, and show empirically that PVar-based filtering leads to more sample-efficient learning. While the idea is interesting, reviewers raised concerns regarding the technical soundness of the method, the strength and breadth of the baselines, and the lack of experimental validation across diverse domains. Unfortunately, these concerns were not adequately addressed during the discussion period (see `Reviewer Concerns`). I therefore recommend rejection of this submission.

**Reviewer Concerns:**

### Reviewer JLpC

* [resolved] Code-domain evaluation
* [resolved] Data subset selection baselines
* [resolved] Ablation and correlation studies

### Reviewer YE9r

* [still remaining] Theoretical overstatements: Although additional theory was introduced, it does not fully address the reviewer’s concerns.
* [still remaining] Baseline comparisons: Performance gains remain marginal.
* [resolved] Hyperparameter search

### Reviewer bZmS

* [resolved] Connection between theory and method: The authors added a new theorem to justify retaining high-PVar samples.
* [resolved] Ablations on sampling strategy and split ratio: Additional experiments were provided.

### Reviewer UBZj

* [resolved] Evaluation on other domains
* [still remaining] Other preference alignment algorithms: Comparisons with SimPO, KTO, and ORPO were added, but improvements are marginal.
* [still remaining] Pair-level response filtering
* [still remaining] Model-induced uncertainty

**Reviewer Scores:**

* Reviewer JLpC: 4 $\rightarrow$ (4 por 6)

* Reviewer YE9r: 4 $\rightarrow$ 4

* Reviewer bZmS: 4 $\rightarrow$ (4 or 6)

* Reviewer UBZj: 4 $\rightarrow$ 4

---

### Decision · Program_Chairs · 2026-01-26

Reject